# Mitochondria serve as a holdout compartment for aggregation-prone proteins hindering efficient degradation

Maria E. Gierisch [1] ✉, Enrica Barchi [1], Mirco Marogna [1], Moritz H. Wallnöfer [1], Maria Ankarcrona [2], Luana Naia [2], Florian A. Salomons[1] & Nico P. Dantuma [1] ✉

The accumulation of protein aggregates has been causatively linked to the pathogenesis of neurodegenerative diseases. Here, we conduct a genome-wide CRISPR-Cas9 screen to identify cellular factors that regulate the degradation of an aggregation-prone reporter. Genes encoding proteins involved in mitochondrial homeostasis, including the translation factor eIF5A, are enriched among suppressors of the degradation of the reporter. Genetic or chemical inhibition of eIF5A leads to dissociation of the aggregation-prone substrate from mitochondria, which is accompanied by enhanced ubiquitin-dependent proteasomal degradation. The presence of an aggregation-prone, amphipathic helix that localizes the reporter to mitochondria is crucial for the stimulatory effect of eIF5A inhibition on proteasomal degradation. Additionally, inhibition of eIF5A also enhances degradation of mutant huntingtin and α-synuclein, two disease-associated proteins that contain amphipathic helices and mislocalize to mitochondria. We propose that mitochondria serve as a holdout compartment for aggregation-prone proteins. Therefore, preventing mitochondrial localization of aggregation-prone proteins may offer a viable therapeutic strategy for reducing disease-associated proteins in neurodegenerative disorders.

Cells are equipped with intricate protein quality control systems that prevent contamination of the intracellular environment by misfolded or otherwise dysfunctional proteins[1]. These systems capture aberrant proteins and either refold them to their native state or target them for degradation. Given that misfolded proteins are inherently toxic due to their propensity to aggregate, their prompt elimination is critical for cell viability[2]. Several idiopathic and familial neurodegenerative diseases are linked to the age-dependent accumulation of aggregation-prone proteins in neurons, indicating that inefficient clearance of these aberrant proteins lies at the core of the cellular pathology[3]. Lowering the steady-state levels of the disease-associated proteins to prevent their accumulation, aggregation and toxicity presents a promising therapeutic approach. While various strategies have been explored to reduce their synthesis[4–7], leveraging the cell's intrinsic disposal mechanisms for misfolded proteins remains an underexplored avenue

The ubiquitin-proteasome system (UPS) is primarily responsible for degrading misfolded proteins in the cytoplasmic and nuclear compartments. The two key players are ubiquitin, a protein modifier that tags proteins for degradation[8], and the proteasome, a multi-subunit complex that degrades ubiquitinated proteins[9]. The UPS is particularly efficient in clearing misfolded proteins before they form insoluble aggregates[10], with the latter being more appropriate substrates for macro-autophagy as this proteolytic system does not rely on unfolding of proteins[11]. While inefficient clearance of misfolded

[1]Department of Cell and Molecular Biology, Karolinska Institutet, Stockholm, Sweden. [2]Department of Neurobiology, Care Sciences and Society, Division of Neurogeriatrics, Center for Alzheimer Research, Karolinska Institutet, Stockholm, Sweden. ✉e-mail: maria.gierisch@ki.se; nico.dantuma@ki.se

proteins by the UPS has been proposed as an explanation for the accumulation of ubiquitin-positive protein aggregates in neurodegenerative disorders, recent insights indicate that UPS dysfunction is not necessarily a common feature in their etiology[12]. For instance, in mouse models of Huntington's disease (HD) and spinocerebellar ataxia type 7, pathogenic aggregation-prone proteins accumulate despite a largely functional UPS[13–15]. The presence of an operational UPS in neurons that are vulnerable to toxic protein aggregates raises the question of whether this protective mechanism can be harnessed to more effectively target disease-associated proteins.

The UPS has gained interest as a potential therapeutic target due to the numerous druggable proteins involved in the execution of this proteolytic pathway. Most efforts have focused on developing drugs that inhibit the UPS, some of which are now clinically used for treating hematological malignancies[16]. However, for neurodegenerative disorders, stimulating UPS activity to degrade aggregation-prone proteins presents a more challenging goal, as drugs typically exert their therapeutic effects through inhibition rather than stimulation of enzymatic activities. Cells possess complex regulatory circuits that can enhance the UPS capacity in response to acute increases in demand, such as during proteotoxic stress[1]. This adaptability suggests that, when unchallenged, the UPS may not operate at its maximal capacity, indicating that pharmaceutical stimulation may be feasible. A few targets for stimulating UPS activity have been identified[17–20], providing proof-of-principle that UPS activity can be enhanced. However, given the enormous complexity of this proteolytic system and the large number of proteins involved, identifying viable drug targets remains a major challenge.

Mislocalization to mitochondria has been reported for several aggregation-prone proteins associated with neurodegenerative diseases. For example, mutant huntingtin[21], α-synuclein[22] and amyloid β-peptide[23] have been found to localize at the mitochondrial compartment in cellular and animal models for neurodegeneration. The tendency of these proteins to bind mitochondria may be intrinsically linked to their hydrophobic nature, which also contributes to their propensity to form insoluble aggregates. While many studies have investigated the role of mislocalized proteins in mitochondrial dysfunction, little is known about the extent to which mitochondrial sequestration affects the cell's ability to clear aggregation-prone proteins. Conceivably, since proteasomal degradation depends on the proximity of substrates to the ubiquitin ligases that target them for degradation, the subcellular localization of aggregation-prone proteins may affect their degradation kinetics and modulate their accumulation in affected cells.

Here, we report on a genome-wide screen to identify cellular factors that regulate the ubiquitin-dependent proteasomal degradation of aggregation-prone proteins. Our findings indicate that biochemical or genetic inhibition of the translation elongation factor eIF5A, a modulator of mitochondrial homeostasis through its ability to promote translation through poly-proline and other specific motifs[24], accelerates the degradation of aggregation-prone proteins that contain amphipathic helices and mislocalize to mitochondria. Our data suggest that localization of aggregation-prone proteins to mitochondria impedes their degradation, as this limits the accessibility to ubiquitinating enzymes involved in protein quality control. We therefore propose that preventing the sequestration of disease-associated proteins in the protective mitochondrial environment can stimulate their degradation and have identified eIF5A inhibition as a potential target for this purpose.

## Results
### Generation of a reporter cell line for a genome-wide CRISPR-Cas9 screen
Designing a genome-wide CRISPR-Cas9 screen to identify regulators of proteasomal degradation of aggregation-prone proteins using a

reporter substrate involves several technical challenges. First, the inherent toxicity of protein aggregates complicates the stable expression of aggregation-prone proteins necessary for a phenotypic screen[2]. Second, the expression levels of these proteins must allow for the detection of both an increase and a decrease in the steady-state levels. Third, changes in steady-state levels should result from altered protein degradation and be distinguishable from changes due to differences in protein synthesis. To address these issues, we utilized the aggregation-prone reporter yellow fluorescent protein (YFP)-CL1, which can be stably expressed in cells without overt signs of toxicity[25]. Under physiological conditions, YFP-CL1 is distributed throughout the cytosol and nucleus at low but detectable levels without forming intracellular aggregates[26]. Hence, YFP-CL1-expressing cell lines can be maintained without aggregate-related toxicity. However, YFP-CL1 accumulates and forms intracellular aggregates when proteasomal degradation is inhibited[27,28]. Aggregation is the consequence of the CL1 degradation signal, which consists of an amphipathic helix that targets the protein for proteasomal degradation and renders the reporter aggregation-prone[29]. Flow cytometric analysis showed that the unrelated proteasome inhibitors epoxomicin and bortezomib, as well as the E1 ubiquitin activase inhibitor TAK-243 induced accumulation of YFP-CL1 in a human melanoma MelJuSo cell line, while administration of the autophagy inhibitor Bafilomycin A1 did not affect the steady-state levels of the reporter (Supplementary Fig. 1a, b), consistent with a prominent role of the UPS in degradation of the reporter substrate.

To distinguish between changes in protein levels due to degradation versus protein synthesis, the stable reference protein histone H2A tagged with tag blue fluorescent protein (tBFP) was expressed from the same transcript using a p2A ribosome-skipping motif, allowing quantification of the turnover of the reporter as the ratio between YFP and tBFP fluorescence (Fig. 1a). The reporter construct was used to generate a MelJuSo cell line stably expressing tBFP-H2A-p2A-YFP-CL1 from an EF1α promoter. Cycloheximide chase experiments confirmed that the YFP-CL1 was rapidly degraded, whereas the tBFP-H2A was long-lived (Fig. 1b, c). Moreover, treatment with the proteasome inhibitor epoxomicin increased the levels of YFP-CL1 but not of the long-lived tBFP-H2A (Fig. 1b). Microscopic analysis of cells treated with epoxomicin or the ubiquitination inhibitor TAK-243 showed increased YFP-CL1 fluorescence and the formation of perinuclear aggregates, confirming its aggregation-prone nature (Supplementary Fig. 1c, d). Flow cytometric analysis of the stable cell line showed reduced or elevated levels of YFP-CL1 upon cycloheximide or epoxomicin treatment, respectively, which could be quantified relative to the tBFP-H2A reference protein (Fig. 1d).

We then designed a pipeline for a fluorescence-activated cell sorting (FACS)-based genome-wide CRISPR-Cas9 screen to identify positive and negative modulators of YFP-CL1 degradation (Fig. 1e). Cas9-expressing reporter cells were transduced with a lentiviral library consisting of the Brunello knockout sgRNA library[30]. This pooled library has genome-wide coverage and consists of four sgRNAs targeting each gene in the human genome. After culturing the cells for 10-18 days, approximately 0.5% of the cells exhibiting the highest or lowest YFP/tBFP ratios were sorted by FACS in two independent experiments, resulting in the YFP-CL1[high] and YFP-CL1[low] populations, respectively (Fig. 1f). Enriched sgRNAs in these populations were identified by next-generation sequencing.

### Identification of proteins stimulating YFP-CL1 degradation
We first analyzed the YFP-CL1[high] populations from both experiments to validate our screening procedure using the well-characterized degradation pathway of this reporter[29,31–33]. Analysis of the YFP-CL1[high] populations using the Model-based Analysis of Genome-wide CRISPR-Cas9 Knockout (MAGeCK) method[34] (Supplementary Data 1) revealed 824 genes enriched in both experiments (Fig. 2a). Gene ontology (GO) annotation showed that the largest GO classes consisted of genes

linked to the insertion of membrane proteins into the endoplasmic reticulum (ER) and genes linked to protein ubiquitination and degradation (Fig. 2b). This is consistent with earlier studies showing that the engineered CL1 degradation signal is an aggregation-prone amphipathic helix that resembles the membrane targeting sequences of tail-anchored proteins[33] and primarily relies on ER-associated ubiquitination enzymes for its degradation[29,31,32]. Consistent with earlier reports[32,33], we found among the enriched genes in the YFP-CL1[high] population the ubiquitin conjugase UBE2G2 together with AUP1, which recruits UBE2G2 to the ER membrane[35], the ER-associated ubiquitin ligase RNF139 (also known as TRC8), and the ubiquitin elongation factor UBE3C, which extends ubiquitin chains with K29-linked ubiquitin[36] (Fig. 2c). Depletion of these ubiquitination enzymes confirmed their role in regulating YFP-CL1 stability (Fig. 2d, and Supplementary Fig. 2a–c). The importance of ER localization of YFP-CL1 and its ubiquitination enzymes for efficient proteasomal degradation was further supported by the enrichment of sgRNAs targeting EMC1, EMC2, EMC6, EMC8, and MMGT1, which are part of the ER membrane complex (EMC) that facilitates membrane insertion of tail-anchored proteins[37] and the ER-associated protein SRP1, which targets proteins to the ER[38].

An alternative pathway for YFP-CL1 degradation involves the BAG6-GET1 chaperone complex[39], which intercepts soluble tail-anchored proteins and either targets them for membrane insertion or presents them to the E3 ligase RNF126 for ubiquitination and subsequent proteasomal degradation[40,41]. While the sgRNAs targeting BAG6 and GET1 were enriched, those targeting RNF126 were not overrepresented in the YFP-CL1[high] population. However, siRNA-depletion confirmed that RNF126 also promotes YFP-CL1 degradation, albeit less prominently than RNF139 (Fig. 2d, and Supplementary Fig. 2d).

While the identification of several expected hits validated the screening procedure, other candidates were also discovered. These included RNF121, a ubiquitin ligase not previously implicated in targeting YFP-CL1 or other tail-anchored proteins for degradation. Given its localization at the cytosolic face of the ER membrane and its role in degrading a voltage-gated sodium channel[42], RNF121 fits the profile of a YFP-CL1-targeting ubiquitin ligase. Immunostaining confirmed a distribution of RNF121 resembling that of ER-associated proteins (Supplementary Fig. 2e), which was further corroborated by subcellular fractionation, where RNF121 was detected in ER fractions (Supplementary Fig. 2f). Depletion of RNF121 increased YFP-CL1 levels comparable to the levels observed upon RNF126 or UBE3C depletion but not to the same extent as RNF139 depletion (Fig. 2d, and Supplementary Fig. 2g, h). Co-depletion of RNF139 + UBE3C, RNF139 + RNF121, and RNF121 + UBE3C caused an increase in YFP-CL1 that matched or exceeded the predicted additive effect of single depletions, suggesting that these ubiquitination enzymes independently target YFP-CL1 for proteasomal degradation (Fig. 2e, and Supplementary Fig. 2i). While co-depletion of RNF126 + RNF139 or RNF126 + UBE3C suggested independent targeting mechanisms, the combined depletion of RNF126 + RNF121 did not have an additive effect, implying that these ligases may act on successive steps towards proteasomal degradation (Fig. 2f, and Supplementary Fig. 2j). This may reflect the reported requirement for re-ubiquitination at the BAG6-RNF126 complex after ubiquitin-dependent ER extraction for efficient degradation of tail-anchored proteins[43].

The presence of several ubiquitin ligases previously implicated in YFP-CL1 degradation within the YFP-CL1[high] population confirmed that our experimental setup can effectively identify modulators of YFP-CL1 degradation. Additionally, we identified RNF121 as an ER-associated ubiquitin ligase involved in the stability of tail-anchored substrates. The identification of several ER-associated ubiquitin ligases that target YFP-CL1 and the absence of ubiquitin ligases from other membranous

organelles support a model where the cytosolic face of the ER is the predominant site for ubiquitination of this aggregation-prone reporter substrate.

## Identification of proteins suppressing YFP-CL1 degradation

We next focused on the sgRNAs enriched in the YFP-CL1[low] populations. Compared to the YFP-CL1[high] data set, a larger number of enriched genes were found in both experiments (1187 genes) (Fig. 3a), but with an overall lower average enrichment score (Supplementary Table 1). The enriched genes included several genes encoding proteins that regulate the exit of proteins from the ER, underscoring the importance of this compartment in facilitating degradation of YFP-CL1 (Fig. 3b). The GO annotation indicated also a link to the mitochondria as several of the largest GO classes shared a connection to mitochondrial homeostasis. Using MitoCarta 3.0 for a more detailed annotation[44], we found that enriched sgRNAs were directed against genes involved in mitochondrial RNA metabolism, translation and import of mitochondrial proteins, as well as genes encoding components of the respiratory chain complexes I, IV and V (Fig. 3c). Notably, the translation factor eIF5A, which is a modulator of mitochondrial homeostasis[24], was the only gene that was strongly enriched in both experiments (Fig. 3d).

We then performed a screen with siRNAs targeting eIF5A and eleven additional candidates that scored the highest positive logarithm fold change (LFC) based on the MAGeCK analysis and were identified by at least two sgRNAs in both experiments. Two more genes that ranked high in the second experiment of the MAGeCK analysis (BOLA3 and COMMD10) were also included. We determined the effect of depletion of these thirteen candidates on the steady-state levels of four different classes of UPS reporter substrates including, in addition to YFP-CL1, the ERAD substrate CD3δ-YFP, the soluble ubiquitin-fusion degradation substrate Ub[G76V]-YFP and the soluble N-end rule substrate Ub-R-YFP[27] (Fig. 3e). Due to the transient nature of siRNA depletion, reporter levels were already determined after 72 h, which is considerably shorter than the 10–18 days after sgRNA transduction employed in the genetic screen. Despite the shorter time span, a >25% reduction in the levels of the YFP reporters was observed upon the depletion of several candidates (Fig. 3f). Interestingly, only the depletion of eIF5A selectively reduced the steady-state levels of YFP-CL1 without a substantial decrease in the levels of ERAD and soluble substrates.

To our surprise, none of the candidates previously identified as suppressors of the UPS were found to be enriched in the YFP-CL1[low] population. To determine whether this was due to the non-saturating conditions of the screen or to the experimental model used, we analyzed the impact of depleting several known suppressors on the degradation of the YFP-CL1 reporter. Notably, siRNA depletion of USP14, UCH-L5 and p38, which have been found to stimulate proteasomal degradation in other experimental models[17,19,20], did not cause a reduction of YFP-CL1 (Supplementary Fig. 3a–c). This suggests that their effects may be substrate- and/or cell-type specific.

## Depletion or inhibition of eIF5A selectively reduces YFP-CL1 levels

Considering the selective effect of eIF5A depletion on YFP-CL1 steady-state levels, we focused our efforts on this candidate. While non-malignant cells primarily express eIF5A, some malignant cells induce expression of its paralog eIF5A2, which is considered an oncogene[45]. Quantitative analysis showed that MelJuSo cells primarily rely on eIF5A with very little expression of eIF5A2 (Supplementary Fig. 4a). Treatment of cells with three independent siRNAs, which target the two reported splice variants of eIF5A efficiently knocked down eIF5A expression (Supplementary Fig. 4b) with very little effect on the very low levels of eIF5A2 (Supplementary Fig. 4c). Importantly, depletion of eIF5A with each of the siRNAs resulted in significantly reduced YFP-CL1

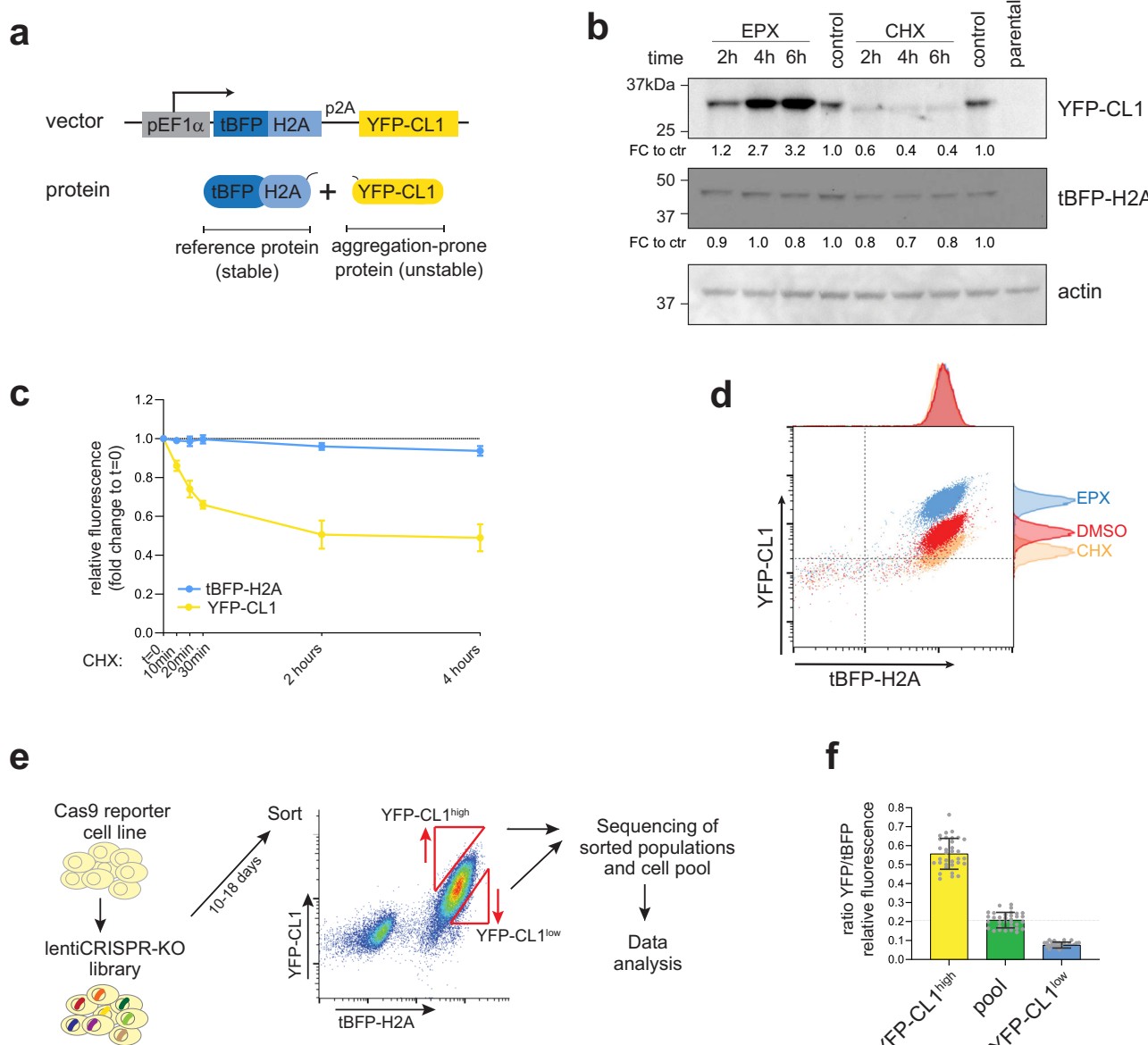

**Fig. 1 | Generation of a reporter cell line for a genome-wide CRISPR-Cas9 screen.**
**a** Schematics of the dual-fluorescent reporter used for the screening. **b** MelJuSo cells stably expressing the dual-fluorescent reporter were treated with 100 nM epoxomicin (EPX) or 20 µg/ml cycloheximide (CHX) for the indicated timings and analyzed by western blotting using an anti-GFP antibody that detects YFP and an anti-tRFP antibody that detects tBFP. Relative band intensities are depicted below the bands. Representative experiment of two. **c** Stable dual-fluorescent reporter cells were cultured for 24 h and treated with 20 µg/ml CHX for the indicated times and analyzed by flow cytometry ($n = 3$, mean ± SD). **d** Stable dual-fluorescent reporter cells were plated for 24 h and treated with 100 nM EPX or 20 µg/ml CHX for the last 7 h and analyzed by flow cytometry. **e** Screening outline, Cas9 dual-fluorescent reporter cells were transduced with a lentiviral CRISPR library at an MOI ~ 0.4 and cultured. The 0.5% populations with the highest (YFP-CL1high) and lowest (YFP-CL1low) YFP-CL1/tBFP-H2A ratios were sorted and subjected for sequencing. **f** Post-sort analysis of the sorted cells when gating for the requirements from **e**. Dots represent the YFP/tBFP ratio at the start of each 2-h long sort, 18 samples for experiment 1 and 15 samples for experiment 2 are pooled, mean ± SD.

levels (Fig. 4a, and Supplementary Fig. 4d) without affecting the soluble substrate UbG76V-YFP (Fig. 4b), confirming the selective effect of eIF5A depletion on the aggregation-prone proteasome substrate.

For its physiological activity in translation, eIF5A strictly depends on a posttranslational modification that converts a specific lysine residue into hypusine, a process mediated by the successive action of deoxyhypusine synthase (DHS) and deoxyhypusine hydroxylase[46]. Since eIF5A and eIF5A2 are the only known hypusination substrates, selective inhibition can be achieved by blocking enzymes involved in hypusination[47]. Indeed, treatment with the DHS inhibitor N1-guanyl-1,7-diaminoheptane (GC7) efficiently prevented eIF5A hypusination

(Supplementary Fig. 4e). Importantly, treatment with GC7 also phenocopied the effect of eIF5A depletion by inducing a dose-dependent reduction in the levels of YFP-CL1 (Fig. 4c). In contrast, the levels of UbG76V-YFP were unaffected and even elevated at the highest GC7 concentration (Fig. 4d). GC7 administration did not affect the expression levels of YFP-CL1 in eIF5A-depleted cells, confirming that the effect of GC7 on YFP-CL1 levels is executed through inhibition of eIF5A hypusination (Fig. 4e, and Supplementary Fig. 4f). Moreover, cell proliferation was only marginally affected by GC7, arguing against the reduced YFP-CL1 levels being a consequence of cellular toxicity (Supplementary Fig. 4g).

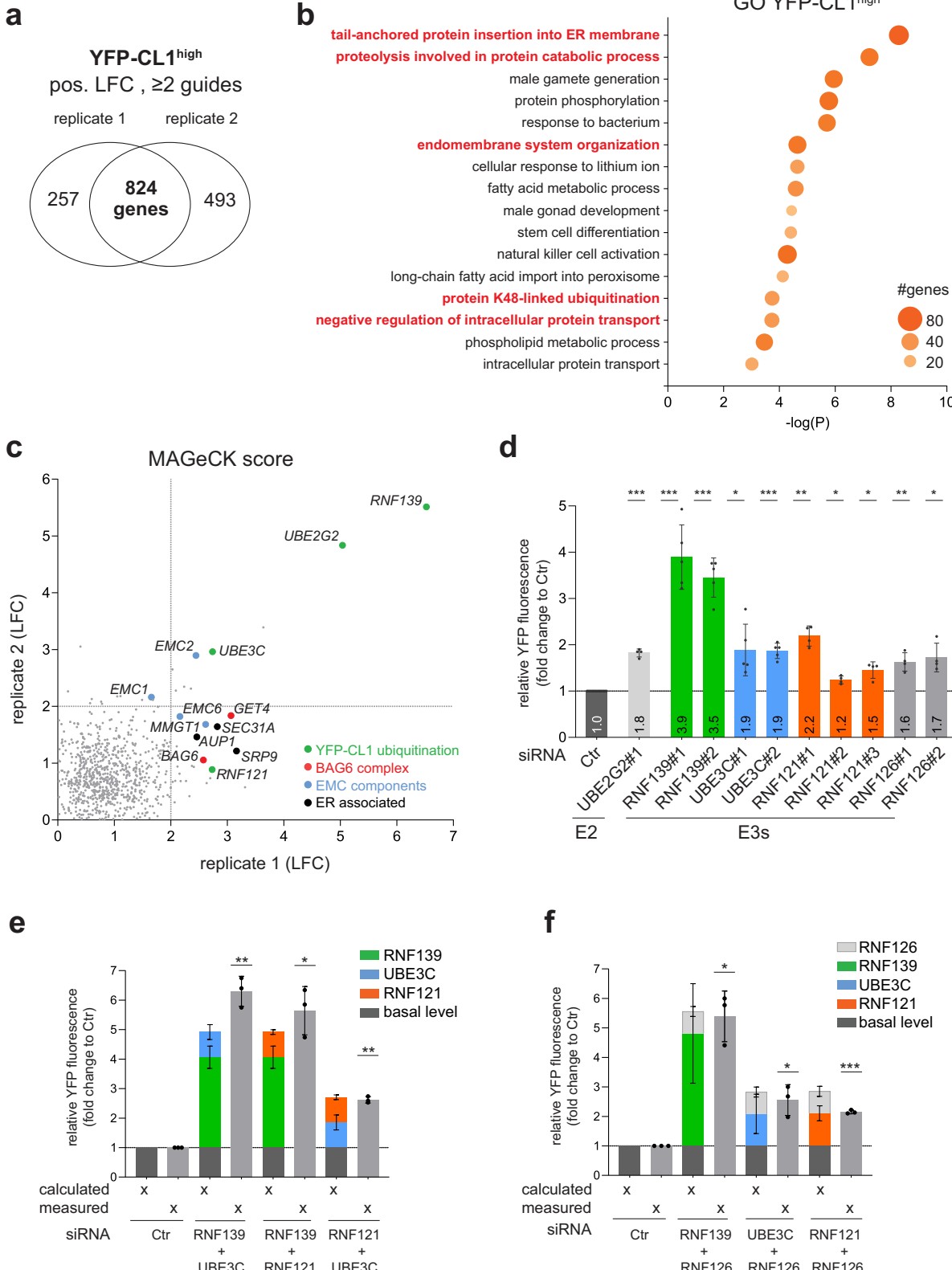

**a**

YFP-CL1$^{high}$
pos. LFC , ≥2 guides

replicate 1    replicate 2

257    **824 genes**    493

**b**

GO YFP-CL1$^{high}$

tail-anchored protein insertion into ER membrane
proteolysis involved in protein catabolic process
male gamete generation
protein phosphorylation
response to bacterium
endomembrane system organization
cellular response to lithium ion
fatty acid metabolic process
male gonad development
stem cell differentiation
natural killer cell activation
long-chain fatty acid import into peroxisome
protein K48-linked ubiquitination
negative regulation of intracellular protein transport
phospholipid metabolic process
intracellular protein transport

#genes
80
40
20

-log(P)

**c**

MAGeCK score

replicate 2 (LFC)

*RNF139*
*UBE2G2*
*EMC2*  *UBE3C*
*EMC1*
*EMC6*  *GET4*
*MMGT1*  *SEC31A*
*AUP1*
*BAG6*  *SRP9*
*RNF121*

YFP-CL1 ubiquitination
BAG6 complex
EMC components
ER associated

replicate 1 (LFC)

**d**

relative YFP fluorescence (fold change to Ctr)

siRNA: Ctr, UBE2G2#1, RNF139#1, RNF139#2, UBE3C#1, UBE3C#2, RNF121#1, RNF121#2, RNF121#3, RNF126#1, RNF126#2

Values: 1.0, 1.8, 3.9, 3.5, 1.9, 1.9, 2.2, 1.2, 1.5, 1.6, 1.7

E2 | E3s

**e**

relative YFP fluorescence (fold change to Ctr)

RNF139
UBE3C
RNF121
basal level

calculated: x x x x
measured: x x x x
siRNA: Ctr | RNF139 + UBE3C | RNF139 + RNF121 | RNF121 + UBE3C

**f**

relative YFP fluorescence (fold change to Ctr)

RNF126
RNF139
UBE3C
RNF121
basal level

calculated: x x x x
measured: x x x x
siRNA: Ctr | RNF139 + RNF126 | UBE3C + RNF126 | RNF121 + RNF126

## Ubiquitin-dependent proteasomal degradation reduces YFP-CL1 levels in eIF5A-deficient cells

The reduction in YFP-CL1 levels induced by eIF5A depletion was rescued by treatment with the proteasome inhibitors epoxomicin or bortezomib, or by blocking ubiquitination with TAK-243, but was not affected by inhibition of autophagy with Bafilomycin A1 (Fig. 5a, and Supplementary Fig. 5a, b). Thus, eIF5A depletion does not impair the synthesis of the YFP-CL1 reporter but promotes its ubiquitin-dependent proteasomal degradation without apparent contribution by autophagy.

We next wanted to identify the ubiquitin ligases that are responsible for the enhanced targeting of YFP-CL1 for proteasomal degradation. As eIF5A deficiency may lead to translational defects, we explored whether ubiquitin ligases involved in ribosome quality

**Fig. 2 | Identification of proteins stimulating in YFP-CL1 degradation. a** SgRNA enrichment of the YFP-CL1^high population in the two independent experiments. **b** Gene Ontology (GO) analysis of 824 genes that overlap between the two experiments generated with Metascape. **c** The log fold change (LFC) of the enriched genes of the two experiments was plotted and selected pathways are color-coded. Source data are provided as a Source Data File. **d** MelJuSo YFP-CL1 reporter cells were transiently transfected with 20 nM siRNAs for 72 h and analyzed by flow cytometry for YFP fluorescence ($n = 5$ for RNF139 and UBE3C, $n = 4$ for RNF121,

RNF126 and UBE2G2, mean ± SD, one-sample $t$-test, $^*P < 0.05$, $^{**}P < 0.01$, $^{***}P < 0.001$). **e**, **f** MelJuSo YFP-CL1 reporter cells were transfected with 10 nM siRNA against single E3 ligases together with 10 nM siCtr or with a combination of two E3 ligase siRNAs (each 10 nM) for 72 h and analyzed by flow cytometry for YFP fluorescence. The calculated bar presents the YFP-CL1 fluorescence of the theoretical combination of the single measured E3 siRNAs while the measured bar presents the actual E3 siRNA combination ($n = 3$, mean ± SD, one-sample $t$-test, $^*P < 0.05$, $^{**}P < 0.01$, $^{***}P < 0.001$).

control (RQC) contributed to YFP-CL1 degradation. However, depletion of LTN1 or ZNF598, two RQC ubiquitin ligases[48], did not prevent the enhanced degradation (Supplementary Fig. 5c, d). Next, we tested the contribution of the cognate ubiquitination enzymes of YFP-CL1 in its enhanced degradation. Depletion of the ER-associated ubiquitin conjugase UBE2G2 prevented the enhanced clearance of YFP-CL1 in GC7-treated cells, indicating an important role for its cognate ubiquitination enzymes (Fig. 5b, and Supplementary Fig. 5e). However, the depletion of the two major E3 ligases RNF139 or UBE3C alone did not prevent the effect of GC7 (Supplementary Fig. 5f). This may be due to the observed redundancy of YFP-CL1 targeting ubiquitin ligases, although we cannot exclude the involvement of other ubiquitin ligases.

Using cell compartment-specific green fluorescent protein (GFP)-CL1 reporters[49], we found that cytosolic localization of the reporter was a prerequisite for enhanced degradation induced by GC7 treatment (Fig. 5c), as the nuclear GFP-CL1 reporter was unresponsive (Fig. 5d), consistent with ubiquitination occurring at the ER. Notably, eIF5A depletion or GC7 treatment did not affect proteasome activity, excluding the possibility that an increase in proteasome capacity is responsible for the accelerated degradation (Supplementary Fig. 5g, h). Together, these findings suggest that the ubiquitination of GFP-CL1 at the cytosolic face of the ER membrane facilitates the enhanced proteasomal degradation in eIF5A-deficient cells.

To further validate the role of eIF5A in regulating YFP-CL1 stability, we deleted the EIF5A gene in the YFP-CL1/tBFP-H2A reporter cell line. Although EIF5A knockout clones were viable, they exhibited reduced growth kinetics (Supplementary Fig. 6a), as has been reported for EIF5A ablation[50]. Quantitative analysis of EIF5A transcripts confirmed efficient ablation of eIF5A with a minor increase observed in the already low levels of the eIF5A2 paralog (Supplementary Fig. 6b). Western blotting showed absence of eIF5A proteins as well as hypusinated proteins in these knockout cells confirming deficiency in overall eIF5A activity (Fig. 5e). In line with the expected accelerated degradation of YFP-CL1, we found that independent EIF5A-knockout clones had on average a 30% reduced YFP-CL1/tBFP-H2A ratio compared to the parental cells (Fig. 5f). The decreased ratio was consistent throughout all clones tested independently of the expression levels of the reporter assessed by the fluorescence intensity of the tBFP-H2A reference protein. Moreover, upon blocking protein synthesis with cycloheximide, the two EIF5A-knockout clones cleared the short-lived YFP-CL1 faster than the parental cells (Supplementary Fig. 6c). The enhanced proteasomal degradation of YFP-CL1 in EIF5A-knockout cells was further supported by the observation that treatment with epoxomicin stabilized YFP-CL1 to a larger extent in EIF5A-knockout cells as compared to the parental cells (Fig. 5g). We conclude that genetic or chemical inhibition of eIF5A accelerates the ubiquitin-dependent proteasomal degradation of YFP-CL1.

### The mitochondrial network regulates YFP-CL1 stability

It has previously been reported that eIF5A inhibition induces a metabolic switch from oxidative phosphorylation to glycolysis, resulting in dramatic changes in the mitochondrial network[24]. Combined with the observation that many mitochondrial proteins were overrepresented among the potential suppressors of YFP-CL1 degradation in our

CRISPR-Cas9 screen, this suggests that the mitochondrial compartment may regulate YFP-CL1 stability.

To better understand the contribution of mitochondria in the turnover of YFP-CL1, we analyzed the subcellular localization of YFP-CL1. Microscopic analysis revealed co-localization of YFP-CL1 with mitochondrial marker Tim50 (Fig. 6a), suggesting that a pool of YFP-CL1 may be associated with the mitochondrial compartment, which was not observed with the soluble Ub^{G76V}-YFP reporter (Supplementary Fig. 7a). Fractionation experiments in which the ER and mitochondrial fractions were separated and probed for the presence of YFP-CL1 confirmed that the reporter is present at the ER and mitochondria, indicating that the amphipathic helical CL1 interacts with both membranous compartments (Fig. 6b). Consistent with the ER being the primary site for YFP-CL1 ubiquitination, we observed YFP-CL1 species migrated at higher molecular weights in a ladder-like pattern typical for ubiquitinated proteins in ER fractions of proteasome inhibitor-treated cells. In sharp contrast, only unmodified YFP-CL1 was detected in the mitochondrial fraction even after proteasome inhibition. The importance of eIF5A for the mitochondrial localization of YFP-CL1 was substantiated by the absence of the reporter in mitochondrial fractions of EIF5A knockout clones (Fig. 6c). Similarly, GC7 treatment reduced reporter levels at mitochondrial fractions although to a less extent than in the knockout cells (Supplementary Fig. 7b). This supports a model where the ER and mitochondrial compartments have opposite effects on YFP-CL1 stability, with the ER compartment playing a central role in ubiquitination and the mitochondrial compartment interfering with this process. Treatment of the mitochondrial fraction with proteinase K revealed that the YFP-CL1 reporter is readily accessible, suggesting that it is not imported into the mitochondrial matrix (Supplementary Fig. 7c). Additionally, we found that positioning of the amphipathic CL1 motif at the N-terminus of YFP (CL1-YFP) dramatically increased co-localization with mitochondria (Supplementary Fig. 7d) and resulted in higher steady-state levels and reduced proteasomal degradation (Supplementary Fig. 7e), consistent with mitochondrial localization hindering efficient clearance of the substrate by the UPS. In this regard, it is noteworthy that the YFP-CL1 aggregates observed in proteasome inhibitor-treated cells often co-localized with mitochondria, unlike the soluble Ub^{G76V}-YFP reporter (Supplementary Fig. 7f).

Consistent with reorganization of the mitochondrial compartment in eIF5A-deficient cells[51], microscopic analysis showed that GC7 treatment altered the mitochondrial distribution to a more fragmented network (Fig. 6d). The mitochondrial network was also changed in the EIF5A knockout clones compared to the parental cell line (Supplementary Fig. 8a, b), confirming the importance of eIF5A for maintaining an elaborate mitochondrial network. While GC7 administration caused a strong reduction in mitochondrial respiration (Supplementary Fig. 8c), the effect of siRNA depletion of eIF5A varied, with only siEIF5A#3 causing a substantial decrease in respiration (Supplementary Fig. 8d). Thus, enhanced degradation of YFP-CL1 correlates with changes in the mitochondrial network that do not have to be accompanied by impaired oxidative respiration. Another compound that affects the integrity of the mitochondrial network is 1-(benzo[d][1,2,3]thiadiazol-6-yl)-3-(3,4-dichlorophenyl)urea (BTdCPU)[52]. Microscopic analysis confirmed that, like GC7 treatment, BTdCPU administration changed the mitochondrial network connectivity

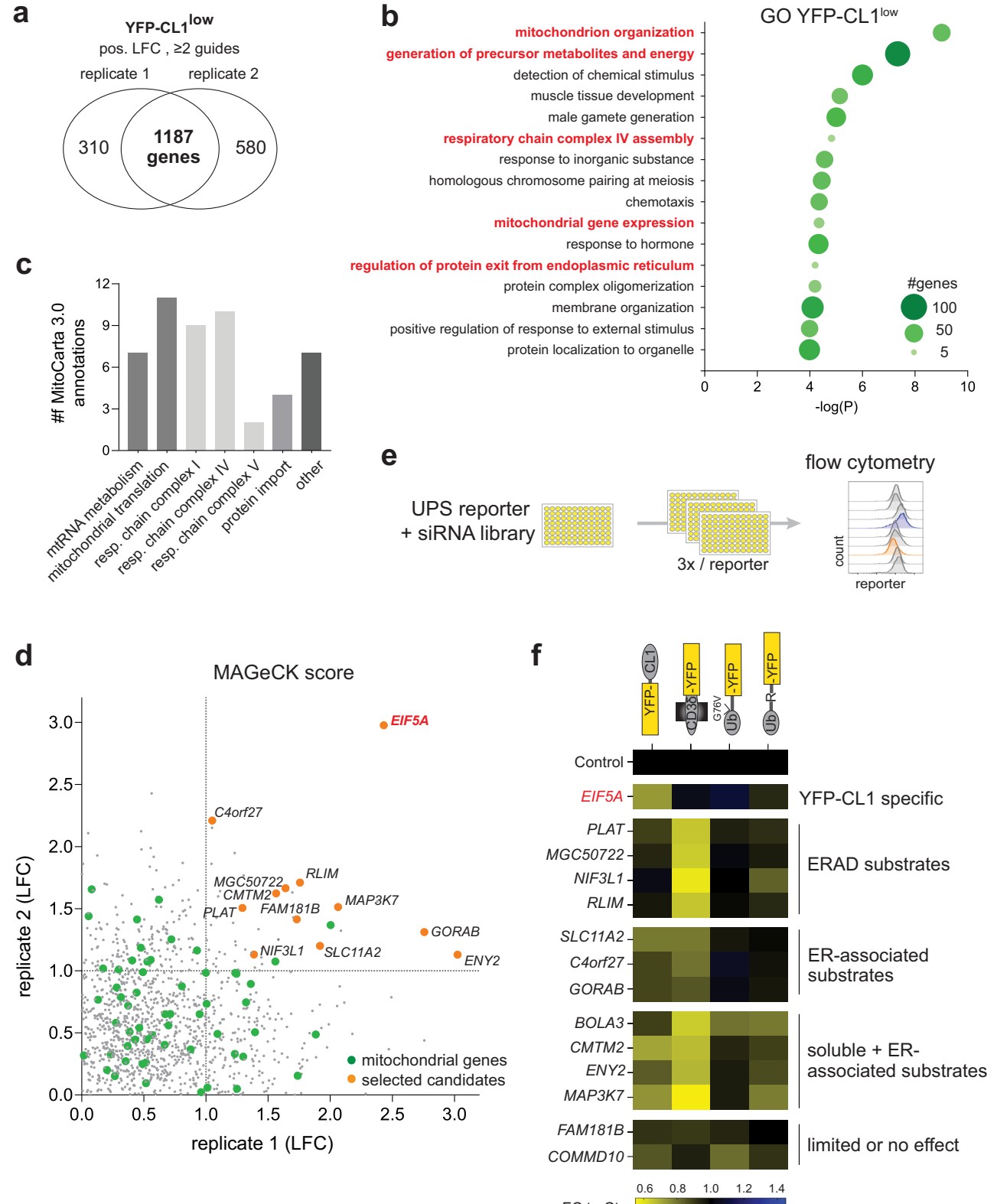

**Fig. 3 | Identification of proteins suppressing of YFP-CL1 degradation. a** SgRNA enrichment of the YFP-CL1^low population over the pool in the two independent experiments with the indicated criteria. **b** Gene Ontology (GO) analysis of 1187 genes that overlap between the two experiments generated with Metascape. **c** MitoCarta 3.0 was used to map the enriched mitochondrial genes to distinct groups. **d** The log fold change (LFC) of the enriched genes of the two experiments was plotted and mitochondrial genes are indicated in green. Top enriched candidates for further evaluation are indicated orange. Source data are provided as a Source Data File. **e** Outline of the siRNA-based mini-screen. **f** MelJuSo reporter cells were transiently transfected with siRNAs against the selected candidates for 72 h and analyzed by flow cytometry for YFP fluorescence. Each square presents pooled data of three independent screening rounds with three different siRNAs calculated as a fold change to the control. Genes are clustered into five different groups.

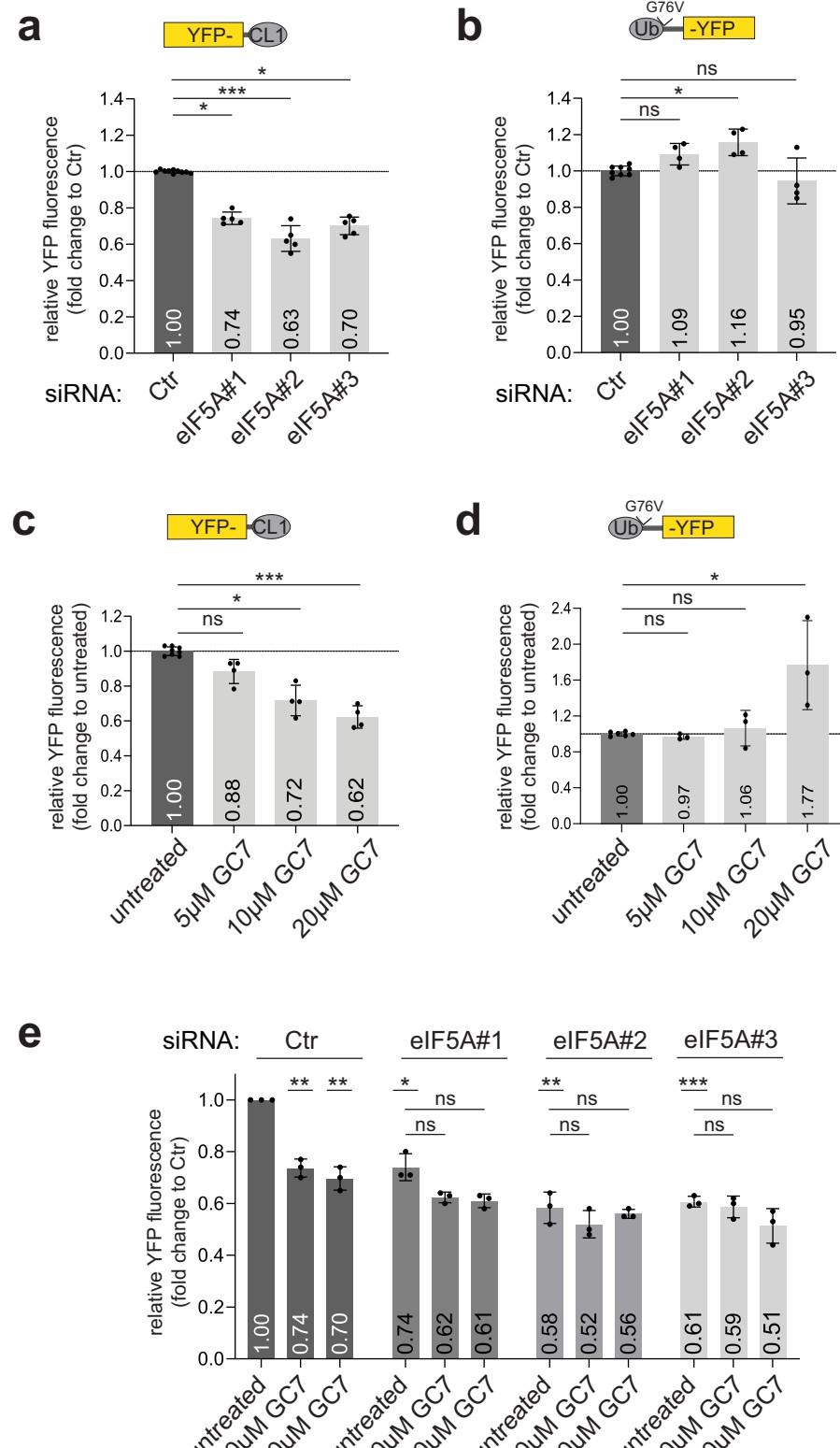

**Fig. 4 | Depletion or inhibition of eIF5A selectively reduces YFP-CL1 levels.**
MelJuSo reporter cells YFP-CL1 (**a**) and Ub$^{G76V}$-YFP (**b**) were transfected with 20 nM
siRNAs against eIF5A for 72 h and analyzed by flow cytometry for YFP fluorescence
($n = 5$ (**a**), $n = 4$ (**b**) with two control samples per independent replicate, mean ± SD,
Kruskal–Wallis test, $^*P < 0.05$, $^{***}P < 0.001$, ns: non-significant). MelJuSo reporter
cells for YFP-CL1 (**c**) and Ub$^{G76V}$-YFP (**d**) were treated with GC7 at the indicated
concentrations for 24 h and analyzed by flow cytometry for YFP fluorescence ($n = 4$

(**c**) and $n = 3$ (**d**) with two control samples per independent replicate, mean ± SD,
Kruskal–Wallis test, $^*P < 0.05$, $^{***}P < 0.001$, ns: non-significant). **e** MelJuSo YFP-CL1
cells were transfected with 20 nM siRNAs against eIF5A for 72 h, treated the last
24 h with 10 or 20 µM GC7 and analyzed by flow cytometry for YFP fluorescence
($n = 3$, mean ± SD, one sample $t$-test when compared to siCtr or Kruskal–Wallis test
when samples were compared within each siEIF5A group, $^*P < 0.05$, $^{**}P < 0.01$,
$^{***}P < 0.001$, ns: non-significant).

(Supplementary Fig. 9a, b). Importantly, treatment with BTdCPU reduced the levels of YFP-CL1 while having the opposite effect on the Ub[G76V]-YFP reporter (Fig. 6e), as observed for GC7 treatment (see Fig. 4c, d). BTdCPU treatment did not affect eIF5A hypusination (Supplementary Fig. 9c, d), which is consistent with the compound altering the mitochondrial network through an independent mechanism[52], and pointing to disruption of the mitochondrial network as the primary cause for YFP-CL1 destabilization. In line with this conclusion, co-treatment with GC7 and BTdCPU had no additional effect on YFP-CL1 stability, indicating that the compounds promote the decrease in YFP-CL1 via a shared mechanism (Fig. 6f). This was further supported by the observation that BTdCPU did not further decrease the YFP-CL1 levels in eIF5A-depleted cells (Supplementary Fig. 9e). It is noteworthy that the effect of GC7 and BTdCPU on YFP-CL1 degradation is not caused by mitochondrial oxidative dysfunction as administration of the mitochondrial complex I inhibitor rotenone did not decrease the steady-state levels of YFP-CL1 (Fig. 6g, and Supplementary Fig. 9f). MitoTracker staining of GC7 and rotenone-treated YFP-CL1 cells confirmed the disparate effects of the compounds on the mitochondrial network (Supplementary Fig. 9g). Together, these data support a model in which eIF5A inhibition enhances the degradation of YFP-CL1 by driving mitochondrial network remodeling, ultimately facilitating release of the substrates from the mitochondria. Given the pivotal role of the ER compartment in YFP-CL1 ubiquitination and the presence of a pool of mitochondria-associated YFP-CL1 that lacks ubiquitin modifications, this suggests that the dissociation of YFP-CL1 from mitochondria, induced by the reorganization of the mitochondrial network in eIF5A-deficient cells, increases the degradation accessible pool of YFP-CL1 and promotes clearance of YFP-CL1.

### The amphipathic helical nature of CL1 is critical for enhanced degradation in eIF5A-deficient cells

We next investigated whether the enhanced ubiquitin-dependent proteasomal degradation of YFP-CL1 in eIF5A-deficient cells depends on the amphipathic nature of the CL1 degradation signal. Previous studies have shown that amino acid substitutions reducing the hydrophobicity of the amphipathic helix in the CL1 domain decrease its ubiquitination by ER-localized ubiquitin ligases[29], and stabilize YFP-CL1[33]. To explore the importance of the amphipathic nature of the CL1 degradation signal in the enhanced degradation induced by eIF5A depletion, we utilized a CL1 mutant with a distorted amphipathic helix[33] (Fig. 7a). It has been previously shown that substitution of these amino acids results in a reduced membrane association in cells[33]. As expected, the amino acid substitutions resulted in increased steady-state levels of the YFP-CL1* (Fig. 7b) and reduced the accumulation of the reporter in response to proteasome inhibition (Fig. 7c), suggesting less efficient targeting for proteasomal degradation. This goes along with a strongly reduced ability of YFP-CL1* to bind mitochondrial membranes compared to YFP-CL1 or CL1-YFP (Fig. 7d). Microscopical analysis also confirmed that the reduced hydrophobicity was accompanied by a decreased tendency to aggregate into intracellular inclusions upon proteasome inhibition (Fig. 7e, f). Importantly, degradation of YFP-CL1* was not enhanced in eIF5A-depleted cells (Fig. 7g), indicating that the presence of an amphipathic helix in the CL1 degradation signal is critical for the constitutive and enhanced degradation of YFP-CL1.

### eIF5A deficiency stimulates degradation of neurodegeneration-associated proteins

Several aggregation-prone proteins linked to neurodegenerative disease contain amphipathic helices and mislocalize to mitochondria[53]. Mutant huntingtin, responsible for HD[54], contains a single amphipathic helix next to the expanded polyglutamine repeat that is responsible for its propensity to aggregate[55]. The N-terminal amphipathic helix has been shown to promote aggregation[56] and facilitate localization to

mitochondria[57]. Indeed, fractionation experiments confirmed the presence of GFP-tagged N-terminal fragment of mutant huntingtin (N-mHtt-GFP) at mitochondria (Supplementary Fig. 10a). Notably, N-mHtt-GFP formed inclusions in the perinuclear region in proximity to mitochondria (Supplementary Fig. 10b). Consistent with a role of mitochondria in stability of huntingtin, administration of GC7 resulted in a decrease in the steady-state levels of N-mHtt-GFP (Fig. 8a). We quantified by flow cytometry the number of cells with N-mHtt-GFP inclusions using PulSA, which detects the presence of GFP aggregates based on decreased width and increased intensity of the GFP signal[58] (Supplementary Fig. 10c). The decrease in N-mHtt-GFP was accompanied by a reduction in the number of cells with N-mHtt-GFP inclusions measured by flow cytometry[58] (Fig. 8b). Using an inducible N-mHtt-GFP cell line, administration of BTdCPU reduced soluble N-mHtt-GFP levels to a similar extent as GC7 treatment (Fig. 8c). A similar reduction in the soluble levels was observed upon depletion of eIF5A, which was partly prevented by administration of proteasome inhibitor (Fig. 8d). Introduction of mutations that distorted the amphipathic helix reduced the propensity of N-mHtt-GFP to aggregate (Supplementary Fig. 10d), consistent with the reported aggregation promoting role of the amphipathic region[56], which was accompanied by a reduced ability of GC7 to reduce N-mHtt-GFP steady-state levels (Fig. 8e). In contrast, GC7 had little or no effect on the steady-state levels of mutant ataxin-1 (Supplementary Fig. 10e), a disease-associated polyglutamine protein that lacks an amphipathic helix. As both mutant ataxin-1 and huntingtin contain polyglutamine expansions and show a similar stabilization upon proteasome inhibition (Supplementary Fig. 10f), this suggests that the presence of an amphipathic helix is an important determinant for the enhanced degradation in the absence of functional eIF5A. In line with the model that eIF5A deficiency results in reduced mitochondrial association of amphipathic helix-containing proteins, western blotting showed that the relative pool of N-mHtt–GFP was strongly reduced in the mitochondrial fraction of *EIF5A* knockout cells as compared with parental cells (Fig. 8f, g).

Another prime example of a disease-associated, amphipathic helix-containing protein is α-synuclein, which is linked to Parkinson's disease and other neurodegenerative diseases[59]. This aggregation-prone protein contains two amphipathic helices[60] and localizes to mitochondrial membranes[53]. To address whether eIF5A depletion can also stimulate the degradation of mutant α-synuclein, we generated a stable cell line that allowed inducible expression of GFP-tagged α-synuclein[A53T]. GFP-α-synuclein[A53T] could be readily detected in the mitochondrial fraction (Supplementary Fig. 10a). Consistent with the proposed effect of eIF5A deficiency on amphipathic helix-containing proteins, depletion of eIF5A reduced GFP-α-synuclein[A53T] levels (Fig. 8h). Moreover, administration of GC7 or BTdCPU also resulted in a decrease in the GFP-α-synuclein[A53T] levels (Supplementary Fig. 10g).

Together, these findings show that preventing mitochondrial localization of aggregation-prone proteins through eIF5A inhibition or depletion can enhance their clearance. Thus, the enhanced degradation of the YFP-CL1 reporter in eIF5A-deficient cells can be extrapolated to disease-associated proteins and, therefore, bears relevance to neurodegenerative disorders.

### Discussion

Despite detailed knowledge of aggregation-prone proteins that cause neurodegenerative disease, little can be done to mitigate the devastating consequences for patients suffering from these conditions. As cells are equipped with protein quality control systems that identify, intercept, and destroy aggregation-prone proteins[1], leveraging proteasomal degradation using the cell's innate protective mechanisms may allow selective targeting of disease-associated proteins while limiting collateral damage. The fact that cells under challenging conditions prevent protein aggregation by enhancing the degradation of

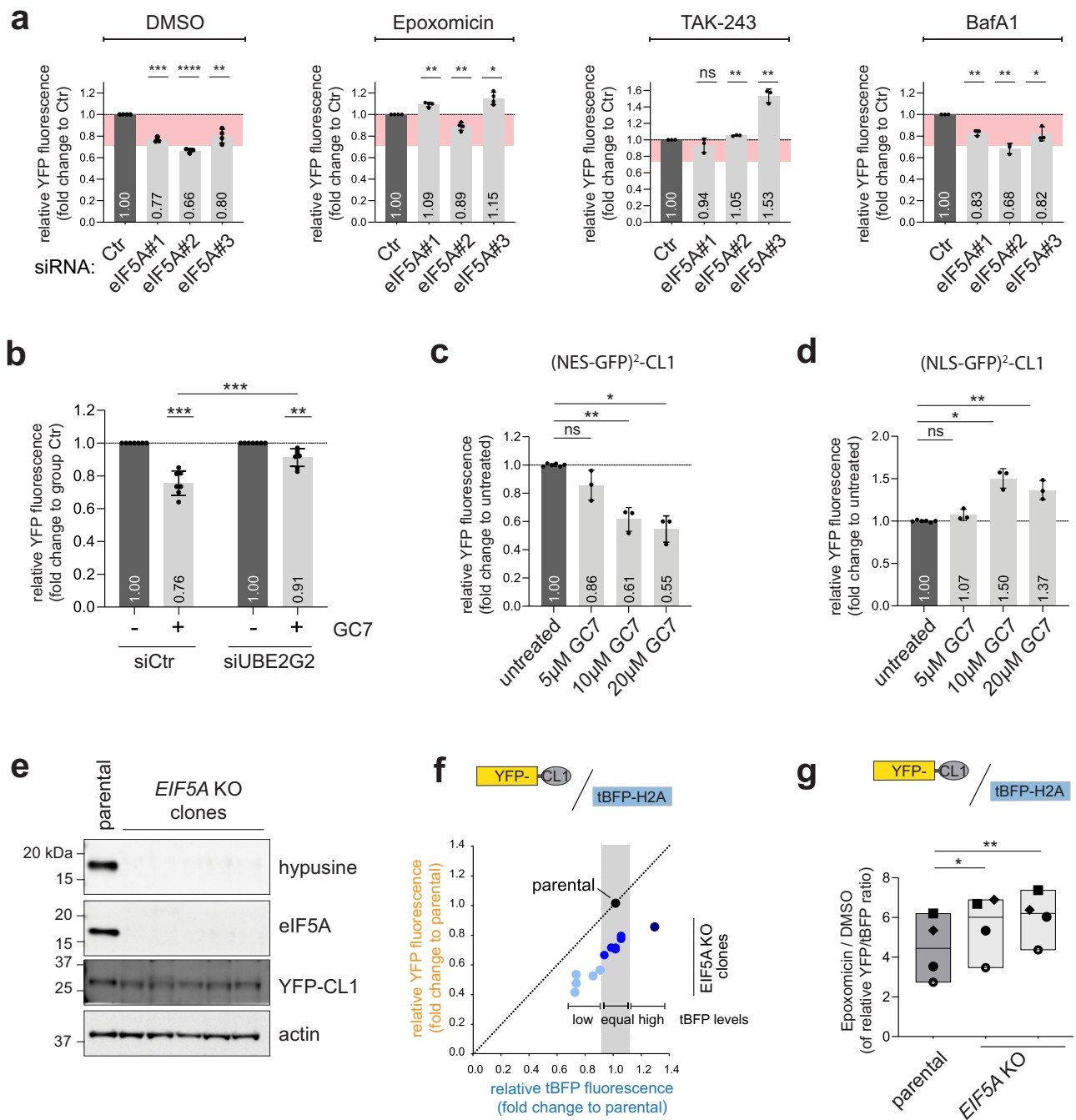

**Fig. 5 | Enhanced ubiquitin-dependent proteasomal degradation reduces YFP-CL1 levels in eIF5A-deficient cells. a** MelJuSo reporter cells expressing YFP-CL1 were transfected with 20 nM siRNAs against eIF5A for 72 h. The final 8 h cells were incubated with 100 nM epoxomicin, 1 μM TAK-243 or 100 nM BafA1, and represented as a fold change to the control siRNA ($n = 4$ for DMSO and EPX, $n = 3$ for TAK-243 and BafA1, mean ± SD, one-sample $t$-test, $^*P < 0.05$, $^{**}P < 0.01$, $^{***}P < 0.001$, $^{****}P < 0.0001$, ns: non-significant). **b** MelJuSo YFP-CL1 cells were transfected with 20 nM siRNA for 72 h and treated the last 24 h with 12 μM GC7. Samples were analyzed by flow cytometry for YFP expression and each siRNA condition was related separately to their respective control ($n = 5$, mean ± SD, unpaired-test comparing GC7 treated conditions and one sample $t$-test comparing to its respective control, $^{**}P < 0.01$, $^{***}P < 0.001$). MelJuSo cells expressing cytosolic (NES-GFP)²-CL1 (**c**) and nuclear (NLS-GFP)²-CL1 (**d**) reporter substrates were treated with indicated concentrations of GC7 for 24 h and samples were analyzed by flow

cytometry for GFP expression ($n = 3$ with two control samples per independent replicate, mean ± SD, Kruskal–Wallis test, $^*P < 0.05$, $^{**}P < 0.01$, ns: non-significant). **e** Single clonal *EIF5A* knockout (*EIF5A* KO) cell lines in MelJuSo tBFP-H2A-p2A-YFP-CL1 reporter background were analyzed for eIF5A depletion by western blotting using indicated antibodies. **f** Twelve clonal *EIF5A* knockout MelJuSo tBFP-H2A-p2A-YFP-CL1 reporter cell lines were analyzed by flow cytometry for YFP and tBFP fluorescence. KO cell lines were plotted for their YFP/tBFP distribution in comparison to parental reporter cells. **g** MelJuSo parental cells and two *EIF5A* KO tBFP-H2A-p2A-YFP-CL1 reporter cell lines with the same tBFP intensities were treated 16 h with 100 nM epoxomicin and analyzed by flow cytometry for their YFP/tBFP ratio. Data are presented as box plots showing the median and the minimum/maximum range, symbols are indicating independent experiments ($n = 4$, repeated measures one-way ANOVA, $^*P < 0.05$, $^{**}P < 0.01$).

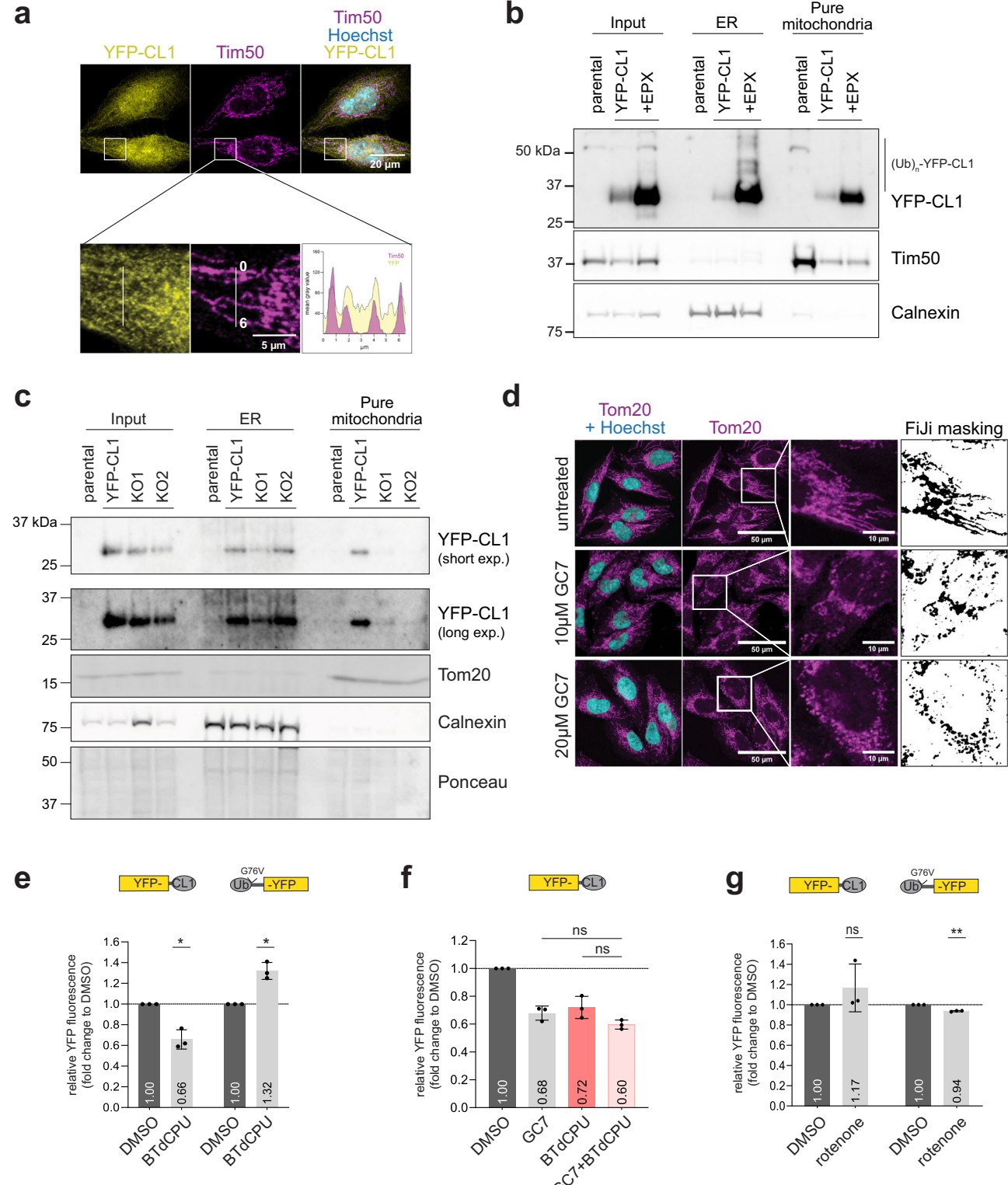

misfolded proteins as part of stress responses implies that it is possible to stimulate the UPS within the boundaries of its physiological capacity[61].

In agreement with earlier studies[32,33], we found that ER localization and ER protein quality control components are the main determinants for efficient proteasomal degradation of the aggregation-prone YFP-CL1 reporter. Consistent with this scenario, sgRNAs targeting genes encoding the insertion or removal of proteins from the ER were enriched in the YFP-CL1[high] and YFP-CL1[low] populations, respectively,

suggesting that increased residence time at the ER enhances degradation of this tail-anchored reporter. Unexpectedly, our CRISPR-Cas9 screen indicated an opposite link between mitochondrial homeostasis and YFP-CL1 degradation, suggesting that mitochondrial localization may hinder efficient clearance. Although the GO analysis showed that sgRNAs targeting genes encoding mitochondrial proteins were most abundant in the sorted population with reduced YFP-CL1 levels, the enrichment was relatively modest, with the striking exception of eIF5A, a translation factor that modulates mitochondrial homeostasis[24].

**Fig. 6 | The mitochondrial network regulates YFP-CL1 stability. a** MelJuSo YFP-CL1 cells were stained for reporter levels using an anti-GFP antibody and for the mitochondrial network using an anti-Tim50 antibody and imaged by confocal microscopy. Co-localization was assessed by FiJi software analysis. Representative images, scale bar = 20 μm. MelJuSo dual-fluorescent reporter cells were fractionated and 2.5 μg (**b**) or 2 μg (**c**) protein of each fraction sample was analyzed by western blotting using anti-GFP for reporter expression and compartment specific antibodies anti-Tim50 or anti-Tom20 and anti-Calnexin. **d** MelJuSo cells were treated with 10 μM and 20 μM GC7 for 24 h, stained for the mitochondrial network with an anti-Tom20 antibody and imaged by confocal microscopy. Representative images, larger image: scale bar = 50 μm, zoom-in image: scale bar = 10 μm. Deconvoluted images, as indicated by a representative cell, were used for staining quantifications. **e** MelJuSo YFP-CL1 or Ub$^{G76V}$-YFP cells were treated with 10 μM BTdCPU and samples were analyzed by flow cytometry for YFP expression ($n = 3$, mean ± SD, one-sample $t$-test, $^*P < 0.05$). **f** MelJuSo YFP-CL1 cells were treated with 10 μM GC7, 10 μM BTdCPU or a combination of both compounds. Samples were analyzed by flow cytometry for YFP expression ($n = 3$, mean ± SD, Kruskal–Wallis test, ns: non-significant). **g** MelJuSo YFP-CL1 or Ub$^{G76V}$-YFP cells were treated with 200 nM rotenone for 24 h and samples were analyzed by flow cytometry for YFP expression ($n = 3$, mean ± SD, one-sample $t$-test, $^{**}P < 0.01$, ns: non-significant).

Although several mechanisms have been proposed for the intricate role of eIF5A in mitochondrial homeostasis, such as its role in synthesis of proteins that contain a mitochondrial localization signal[62], or proteins that are involved in mitochondrial import[63], the molecular mechanism is presently poorly understood. Our further analysis confirmed that genetic or chemical interference with eIF5A function accelerates the degradation of both engineered and disease-associated, aggregation-prone proteins.

The tendency of the aggregation-prone CL1 domain to interact with membranous structures, particularly the ER and mitochondria, appears to be critical for the effect of eIF5A on the proteasomal degradation of this protein. Even though the CL1 motif was originally identified as an ER-binding peptide[29], it also resembles the amphipathic helical motifs that target tail-anchored proteins to other membranous organelles, including mitochondria[64]. Our data suggest that YFP-CL1 associates with the mitochondrial outer membrane and is not subject to import, similar to what has been shown for the ER[29]. We propose that localization of the reporter at the mitochondria hinders proteasomal degradation by keeping it out of reach of the primary systems involved in protein quality control, such as those located at the cytosolic face of the ER (Fig. 9). Interestingly, recent studies revealed a role for the ER-associated ubiquitin ligase HRD1 in degradation of outer mitochondrial membrane proteins[65,66]. Thus, YFP-CL1 and other aggregation-prone proteins may be subject to similar protein quality control mechanisms that apply to mitochondrial proteins.

Due to the high levels of protein synthesis at the ER in combination with the challenging environment for protein folding, the ER requires an efficient system for dealing with misfolded proteins, as evidenced by the presence of various sensors that mobilize the unfolded protein response[67]. The presence of this efficient system may explain why the cytosolic face of the ER may function at the same time as a local hub for dealing with cytosolic aggregation-prone proteins that, due to the intrinsic nature of these domains, also tend to associate with ER membrane. Even though localizing the ubiquitination of aggregation-prone proteins at the ER may be beneficial in terms of efficiency, this may also impose limitations, as mislocalization to other membranous organelles may provide an escape route.

Whether the CL1 motif interacts passively with the outer mitochondrial membrane due to its hydrophobic nature or engages the mitochondrial machinery for inserting tail-anchored proteins remains to be determined. Although the molecular mechanism responsible for the reduced association of YFP-CL1 with mitochondria in eIF5A-deficient cells remains to be elucidated, our data suggest that a general impairment of mitochondrial function is unlikely to be the explanation, given that inhibition of the mitochondrial complex I by rotenone did not accelerate the degradation of the reporter. Conceivably, regulatory mechanisms that inhibit the insertion or stimulate the extraction of tail-anchored mitochondrial proteins may also be involved in the reduced mitochondria association and enhanced degradation in the absence of functional eIF5A. It is noteworthy that a recent study revealed a role for eIF5A in regulating mitochondrial import in budding yeast through facilitating translation of the TIM50 translocase[63]. However, the proline-rich sequence that is responsible for the eIF5A-dependent translation of yeast TIM50 is not conserved in its human orthologue (TIM50_HUMAN; UniProt ID: Q3ZCQ8).

Recent studies suggest that eIF5A is also involved in other processes related to protein quality control, suggesting a more integrated role of this translation factor in maintaining protein homeostasis. Notably, eIF5A is critical for CAT tailing[68], an important step in targeting newly synthesized polypeptides for ubiquitin-dependent proteasomal degradation through RQC[69]. It has also been reported to have a stimulatory effect on autophagy by facilitating efficient translation of Atg3[70]. However, it may not be confined to Atg3 expression as it also promotes translation of TFEB, resulting in increased mitophagy dependent on Atg3, Atg7 and PINK[71–73]. However, despite the role of eIF5A in RQC and autophagy, neither was found to be involved in the accelerated degradation of YFP-CL1. Instead, degradation appeared to be facilitated by its cognate ubiquitin conjugase. This suggests that changes in the levels of hypusinated eIF5A can potentially rewire proteolytic systems in response to the metabolic state of cells.

Our finding that eIF5A affects not only the synthesis but also the degradation of proteins adds another level of complexity to its regulatory role in the cellular proteome. A major challenge in deciphering the effect of eIF5A on the cellular proteome will be to distinguish if reduced steady-state levels of a protein are caused by inhibition of translation, accelerated degradation, or a combination of both. In this context, it is intriguing that even though the reduction in Atg3 levels has been attributed to the presence of a non-canonical tripeptide motif that requires eIF5A for efficient translation[70], it also contains an amphipathic helix important for its interaction with autophagosomes[74]. Therefore, it remains possible that in addition to the effect on translation, the reduced Atg3 levels may also result from accelerated proteasomal degradation in the absence of hypusinated eIF5A.

The question remains whether eIF5A could be a viable therapeutic target to accelerate UPS-mediated clearance of disease-associated proteins in neurodegenerative diseases. The unique dependency of eIF5A on hypusination may provide a promising avenue to chemically fine-tune its activity and accelerate clearance of aggregation-prone proteins. Encouragingly, short-term systemic inhibition of eIF5A hypusination by GC7 has shown protective effects in mouse models for type 2 diabetes[75], renal ischemia[51], and stroke[76]. At the same time, the reduction in autophagy and changes in mitochondrial homeostasis in response to inhibition of eIF5A hypusination[70,72,73] may be less desirable from a therapeutic perspective in these diseases, given the important role of autophagy in proteostasis. However, inhibition of eIF5A may not be the only means to reach this goal, as alternative ways to interfere with the mitochondrial localization of disease-associated proteins, without affecting mitochondrial homeostasis, may be equally effective in reducing the levels of these proteins and mitigating their toxic effects.

## Methods

### Cell culture and reagents

MelJuSo cells (RRID:CVCL1403, gift from Jacques Neefjes, The Netherlands Cancer Institute) and MelJuSo-derived stable clonal cell lines were cultured in DMEM with GlutaMAX (Invitrogen) and

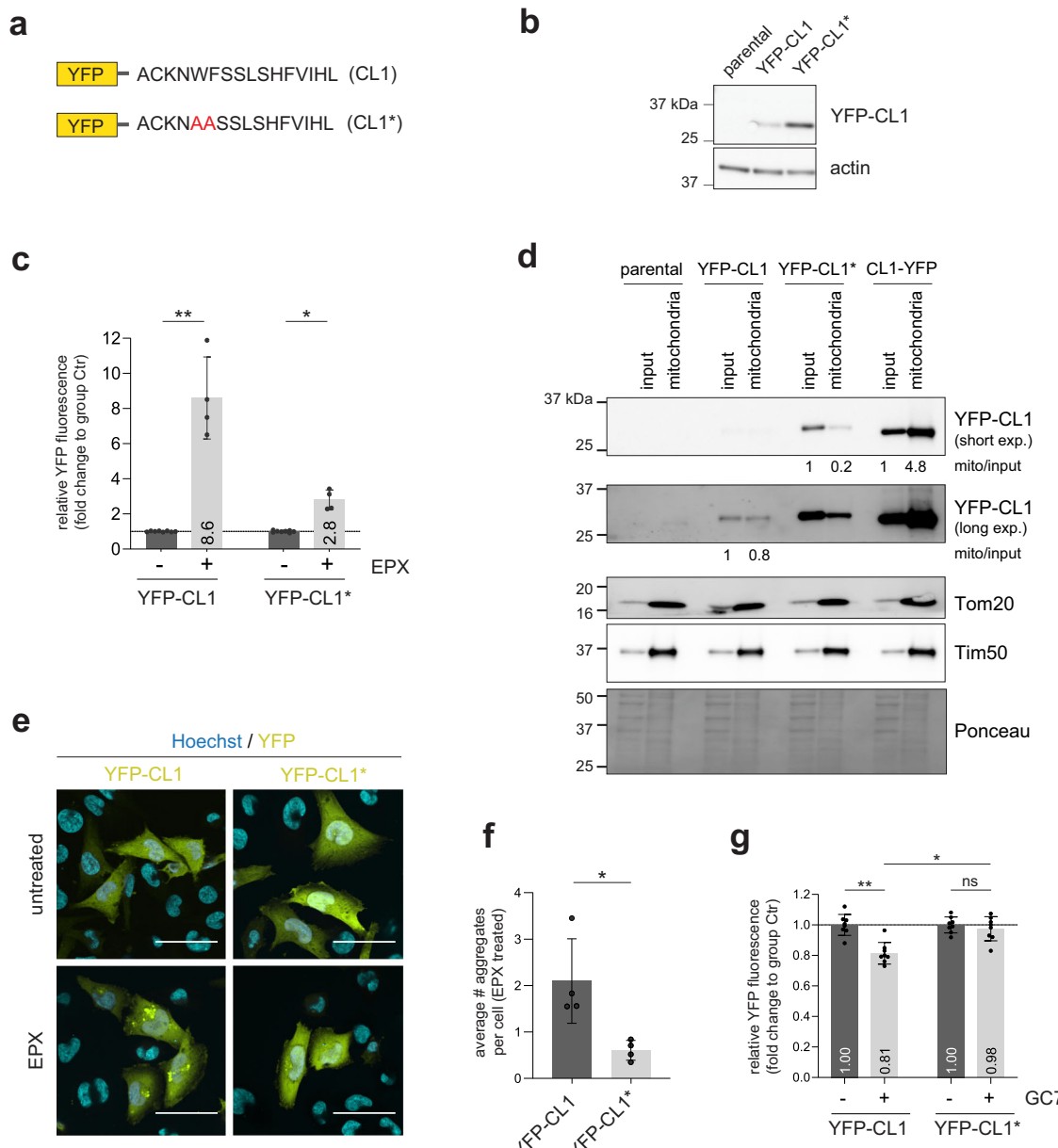

**Fig. 7 | The amphipathic helical nature of CL1 is critical for enhanced degradation in eIF5A-deficient cells. a** Schematics of the CL1 and CL1* sequences. **b** YFP-CL1 and YFP-CL1* were transiently overexpressed in MelJuSo cells for 24 h and analyzed by western blotting using an anti-GFP antibody. **c** YFP-CL1 and YFP-CL1* were transiently overexpressed in MelJuSo cells for 48 h and treated the last 16 h with 100 nM epoxomicin (EPX). Samples were analyzed by flow cytometry for YFP expression separately for each mutant. EPX treated samples were set in relation to their respective control ($n = 4$ with two samples for untreated per independent replicate, mean ± SD, Kruskal–Wallis test, *$P < 0.05$, **$P < 0.01$). **d** YFP-CL1, YFP-CL1* and CL1-YFP were transiently overexpressed in MelJuSo cells for 48 h, fractionated and 5 µg protein of each fraction sample was analyzed by western blotting using anti-GFP antibody for reporter expression and compartment specific antibodies anti-Tim50 and anti-Tom20. **e** YFP-CL1 and YFP-CL1* were overexpressed in MelJuSo cells for 48 h, which were treated the last 16 h with or without 100 nM EPX. Cells were imaged by confocal microscopy. Representative images, scale bar = 50 µm. **f** Images were analyzed for the average number of cells with aggregates per condition ($n = 4$, mean ± SD, Mann-Whitney test, *$P < 0.05$). **g** YFP-CL1 and YFP-CL1* were transiently overexpressed in MelJuSo cells for 48 h, which were treated the last 24 h with 12 µM GC7. Samples were analyzed by flow cytometry for YFP expression separately for each mutant and set in relation to their respective control ($n = 4$ with two samples for untreated and two samples for GC7 treatments per independent replicate, mean ± SD, Kruskal–Wallis test, *$P < 0.05$, **$P < 0.01$, ns: non-significant). Controls are the same as shown in Fig. 7c.

supplemented with 10% fetal bovine serum (Invitrogen) at 37 °C and 5% $CO_2$. MelJuSo reporter cells YFP-CL1, Ub$^{G76V}$-YFP, Ub-R-YFP and CD3δ-YFP have been described previously[27]. Cells were routinely tested for mycoplasma infection. MelJuSo cells have been authenticated by STR profiling 2024/09 (Microsynth). All reagents used in this study can be found in Supplementary Table 2. The tet-off stable cell line was cultured in the presence of 0.1 µg/ml doxycycline (Sigma). Where indicated, cells have been treated with the compounds epoxomicin

(Sigma), bortezomib (Sigma), GC7 (Sigma), BafA1 (Enzo), cyclohex-imide (Sigma), TAK-243 (MedChem Express), rotenone (Sigma), cycloheximide (Sigma) and BTdCPU (MedChem Express) for the indicated time points in antibiotic-free medium.

### Generation of stable cell lines
Stable cell lines were generated by using selection medium 48 h post transfection for 1 week (1.5 mg/ml for G418 (Invitrogen) or 1 µg/ml

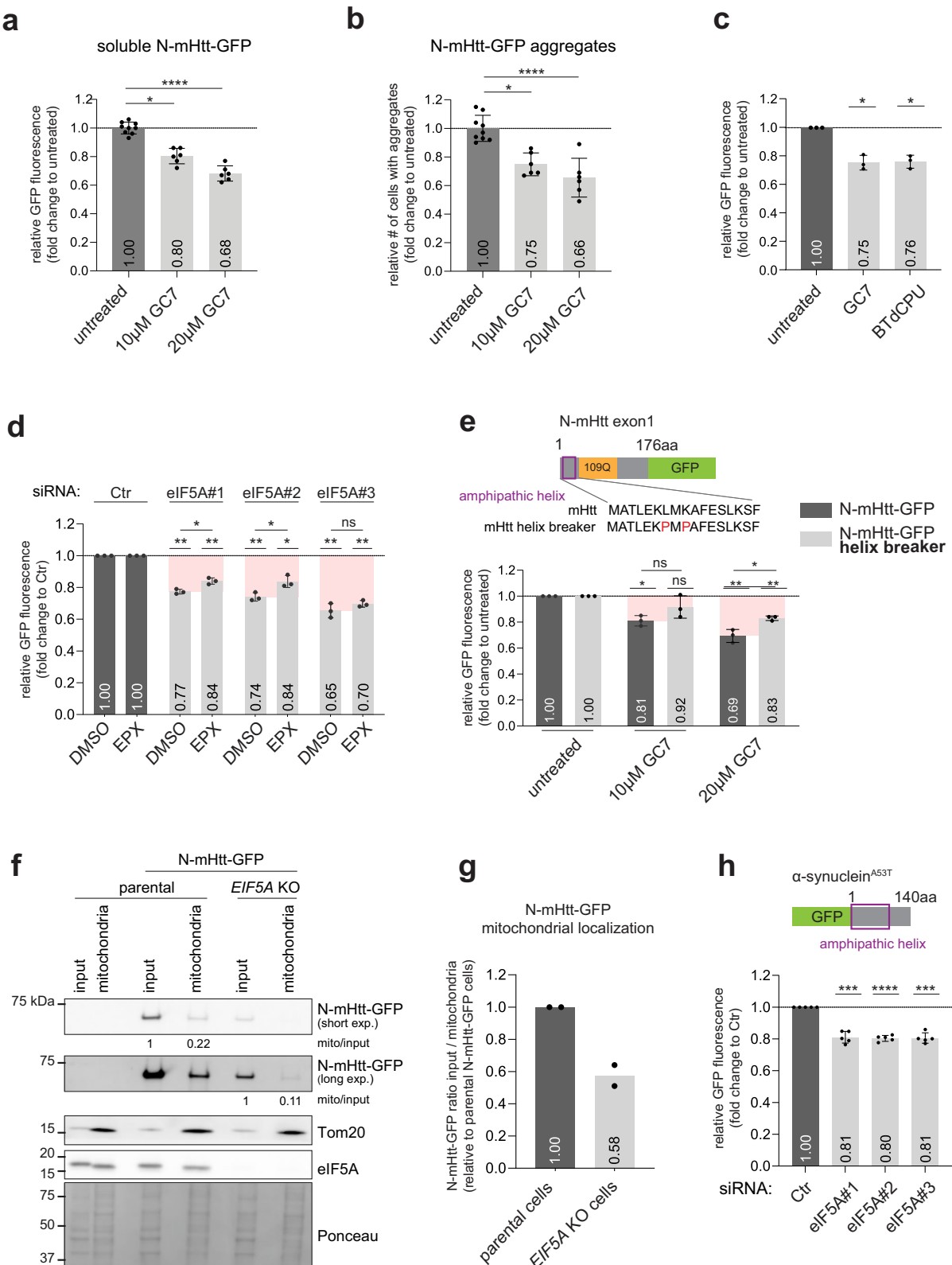

puromycin (Sigma)). Then, cells were either sorted for positive fluorescent markers as a polyclonal cell line or seeded for single colonies, both in the presence of selection medium if needed for an additional 2 weeks. For the screening cell line, the UPS reporter cells were first generated as a monoclonal cell line, and expression was validated by western blotting and flow cytometry. Cells were transduced with Cas9 and selected with blastidicin. To generate *EIF5A* knockout cell lines, the

reporter cell line with stable Cas9 expression was transfected with Lipofectamine™ CRISPRMAX™ Cas9 transfection reagent. In brief, the transfection reagent was mixed with the TRUE synthetic gRNA against *EIF5A* (CRISPR288707_SGM, Invitrogen) in OptiMEM (Invitrogen) and added to the cells for 48 h. Next, cells were grown as single colonies and analyzed by western blotting. To generate inducible cell lines, cells were generated as described above. Medium with 0.1 ng/µl

**Fig. 8 | eIF5A deficiency stimulates degradation of neurodegeneration-associated proteins. a** N-mHtt-GFP was transiently overexpressed in MelJuSo cells for 48 h and treated the last 24 h with 10 μM and 20 μM GC7. GFP expression was analyzed by flow cytometry ($n = 3$) with three samples for untreated and two samples for GC7 treatments per independent replicate, mean ± SD, Kruskal–Wallis test, **$P < 0.01$, ***$P < 0.0001$. **b** Quantification of N-mHtt-GFP aggregates from **a** using PulSa ($n = 3$ with three sample for untreated and two sample for GC7 treatments per independent replicate, mean ± SD, Kruskal–Wallis test, *$P < 0.05$, ***$P < 0.001$). **c** MelJuSo cells with inducible expression of N-mHtt-GFP were plated for 48 h, treated the last 24 h with 10 μM GC7 or 5 μM BTdCPU and analyzed by flow cytometry for GFP expression, ($n = 3$, mean ± SD, one-sample $t$-test, *$P < 0.05$). **d** MelJuSo cells with inducible expression of N-mHtt-GFP were transfected with 20 nM siRNAs for 72 h and the final 16 h cells incubated with 500 nM epoxomicin. GFP expression was analyzed by flow cytometry ($n = 3$, mean ± SD, unpaired $t$-test, *$P < 0.05$, ns: non-significant). **e** N-mHtt-GFP and a helix breaker mutant were transiently overexpressed in MelJuSo cells for 48 h and treated the last 24 h with 10 μM and 20 μM GC7. Soluble GFP expression was analyzed by flow cytometry ($n = 3$, mean ± SD, unpaired $t$-test when compared between the GC7 conditions and one sample $t$-test when compared to each respective untreated, *$P < 0.05$, **$P < 0.01$, ns: non-significant). **f** N-mHtt-GFP was transiently overexpressed in MelJuSo cells for 48 h, fractionated and 4 μg protein of each sample was analyzed by western blotting using anti-GFP antibody and a compartment specific antibody anti-Tom20. The ratio of N-mHtt-GFP at mitochondria to the input was determined in each cell line. Representative experiment of two. **g** Quantification of the ratio mitochondria/input from **f** as fold-change to the N-mHtt-GFP expressing parental cells, $n = 2$. **h** MelJuSo cells with inducible expression of GFP-α-synuclein$^{A53T}$ were transfected with 20 nM siRNAs for 72 h. GFP expression was analyzed by flow cytometry ($n = 5$, mean ± SD, one-sample $t$-test, ***$P < 0.001$, ****$P < 0.0001$).

doxycycline was changed 6 h post transfection and changed every other day. To sort a polyclonal positive cell line, cells were cultured in medium for 14 h, sorted and cultured in the presence of doxycycline.

### Guide library

The genome-wide Brunello sgRNA library[30] was synthesized as 79 bp long oligos (CustomArray, Genscript), double-stranded using a reverse primer (Ultramer_RSL) containing random sequence labels[77] and amplified with primers ds_fw and ds_rev. The insert was gel-purified and cloned by Gibson assembly into pLenti-Puro-AU-flip-3xBsmBI (Addgene #196709), digested with BsmBI. The cloned guide cassettes were amplified by PCR and sequenced by Next Generation Sequencing (NGS) to confirm guide representation. Packaging into lentivirus was performed in using HEK293T (ATCC) and plasmids psPAX2 (Addgene #12260, a gift from Didier Trono) and pCMV-VSV-G (Addgene #8454, a gift from Bob Weinberg). Primer sequences can be found in Supplementary Table 2.

### CRISPR-Cas9 screen

MelJuSo reporter cells were transduced with the Brunello library by lentivirus as described previously[78]. In brief, cells were transduced with a multiplicity of infection of 0.4 and about 1000 cells/guide in 2 μg/ml polybrene. After selection with 2 μg/ml puromycin at days 2–7, cells were further cultured, and a 1000-fold library coverage was maintained. Two independent transductions were performed. Transduced cells were used for downstream FACS at days 10, 11, 16 and 17 (experiment 1) and days 14, 15 and 16 (experiment 2). Cells of the YFP-CL1 fractions and the pools were washed once in phosphate-buffered saline (PBS), and genomic DNA was isolated using the QIAamp DNA Blood Maxi kit (Qiagen). The guide sequences were amplified by PCR, sequencing was performed on an Illumina HiSeq instrument, and the NGS data were analyzed with the MAGeCK algorithm[34]. The MAGeCK analyzed gene lists for experiments 1 and 2 can be found in the Supplementary Data File.

### Genomic DNA, library preparation and NGS sequencing

Genomic DNA was isolated using the QIAamp DNA Blood Mini (Qiagen), and guide and RSL sequences were amplified by PCR as described[77], but with modified primers PCR2_fw and PCR3_fw. The amplicon was sequenced on Illumina HiSeq reading 20 cycles, Read 1 with custom primer CRISPRSeq; 10 cycles index read i7 to read the RSL (not used in the data analysis), and six cycles index read i5 for the sample barcode. NGS data were analyzed with the MAGeCK software, v.0.5.6[34].

### Screen analysis and GO enrichment

To visualize the overlap between the two experiments ("hit list"), we depict genes with a positive MAGeCK score, a minimum of two out of four possible gRNAs in each data set. The overlap between the two datasets has been visualized using Venny 2.1 (Oliveros, J.C. (2007-2015) Venny. An interactive tool for comparing lists with Venn's diagrams. (https://bioinfogp.cnb.csic.es/tools/venny/index.html)).

To identify top hit candidates for selected experimental analysis in the YFP-CL1$^{high}$ conditions, we selected the highest scoring E2 conjugase and E3 ligases with a positive LFC in both experiments and a $p$-value above 0.05. For the YFP-CL1$^{low}$ conditions, top hit candidates with a LFC > 1 and a $p$-value above 0.05 in both experiments were considered and selected for further validation when three individual siRNAs for validation were available.

To analyze pathway enrichment, Metascape[79] was used to analyze gene ontology terms of the selected YFP-CL1$^{high}$ and YFP-CL1$^{low}$ hit lists. The top 16 GO terms were presented as bubble plots over the log of the $P$-value calculated by Metascape[79].

### Flow cytometry and fluorescence-assisted cell sorting

Cells were harvested, washed in PBS and kept at 4 °C until flow cytometry analysis using the BD Canto II or the BD LSR III (both Becton, Dickinson & Company) instruments. Depending on the type of experiment, 20,000–100,000 cells were analyzed. Data were analyzed by FlowJo v10 (Becton, Dickinson & Company). For sorting, cells were additionally washed once in PBS and filtered through a 0.35 μm cell strainer. The sorters BD Aria III and BD Fusion (both Becton, Dickinson & Company) were used. Sorted cells were either collected in PBS or medium and further used for downstream analysis or stable cell line generation. A gating strategy is presented in Supplementary Fig. 12.

### Plasmids and cloning

For side-directed mutagenesis, the QuikChange II kit (Agilent) was used according to the manufacturer's instructions and mutagenesis primers can be found in Supplementary Table 2. NEBuilder assembly (New England Biolabs) was used to generate the double fluorescent screening reporter plasmid by assembling fragments with overhangs generated by PCR of cerulean-histone 2A into the pCMV-YFP-CL1 plasmid digested with Age1 (New England Biolabs). The p2A sequence was produced as an oligo with overhangs for the assembly (Integrated DNA Technologies). From the newly generated plasmid, the pCMV-cerulean sequence was cut out by a double digest of Ase1 and Sca1 (New England Biolabs) and replaced by PCR fragments of pEF1α promoter and tBFP. The digested plasmid and the two fragments were again assembled using NEBuilder and verified by sequencing. The tet-off plasmid was generated by PCR amplifying parts of the backbone of the pCW57.1-MAT2A tet-off plasmid (Addgene #100521), and a puromycin resistance cassette. The tet-off N-mHtt-GFP plasmid was generated by PCR amplifying N-mHtt-GFP from the Addgene #111730 plasmid and a digested backbone of the tet-off-puro plasmid. The tet-off GFP-α-synuclein$^{A53T}$ plasmid was generated by PCR amplifying α-synuclein$^{A53T}$ from the Addgene #40823 plasmid and a digested backbone of the tet-off-puro plasmid. Assembly of

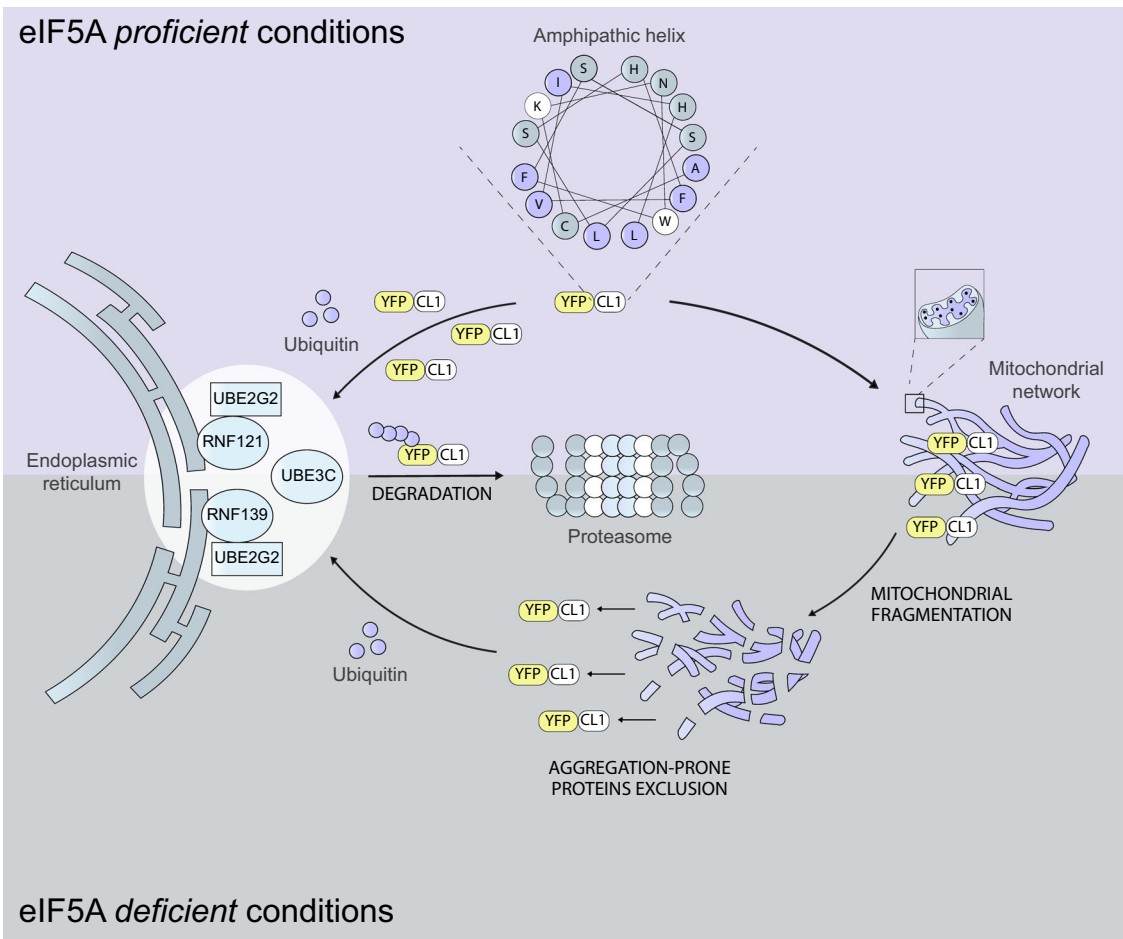

**Fig. 9 | Schematic representation of model.** Under steady-state conditions the aggregation-prone YFP-CL1 is either constantly ubiquitinated by several ER-resident E3 ligases and degraded by the proteasome or mistargeted to mitochondria resulting in a long-lived protein pool. Mitochondrial fragmentation caused by eIF5A depletion or inhibition reduces mislocalization of YFP-CL1 and increases the reporter pool that is susceptible to ubiquitinated and degradation.

fragments was carried out using NEBuilder and verified by sequencing. All plasmids and primers used in this study can be found in Supplementary Table 2.

### Western blotting

Cells were harvested, washed once in PBS and lysed in either 2× LDS buffer (4× LDS sample buffer, 1× protease inhibitor, 1× NuPage reducing agent, 10 μM MG132, 5 mM NEM in PBS) or RIPA buffer (150 mM NaCl, 50 mM Tris, 1% NP40, 0.1% SDS, 0.5% Na-deoxycholate, 10 μM MG132, 5 mM NEM) and for the latter cleared by centrifugation ($20,000 \times g$) and protein concentration was measured by BCA (Invitrogen). Equal protein amounts were loaded. Proteins were separated on a 12% or 4–12% BisTris NuPage precast gel (Invitrogen) and transferred to either nitrocellulose or PDVF membranes (both Invitrogen). Membranes were blocked with 5% milk in TBS-Tween 0.1% for 1 h at room temperature (RT). Primary antibodies were incubated for 2 h at RT or overnight at 4 °C followed by 2 h of 1:10,000 diluted secondary antibody at RT with an IRDye (LI-COR) or HRP-linked (CellSignaling) secondary antibody. A primary antibody list can be found at Supplementary Table 2. Protein detection was carried out either by infrared detection using an Odyssey (LI-COR) or ChemiDoc (BioRad) or using ECL (Cytivia) or SuperSignal™ West Femto Maximum Sensitivity Substrate (Invitrogen) with chemiluminescence detection at the ChemiDoc or using X-ray films (Fuji). Uncropped blots are presented in Source Data File 1 (Main Figures) or in Supplementary Fig. 11 (Supplementary Figures).

### Mitochondrial and ER fractionations

Fractionations have been carried out as previously described[80], and adapted to MelJuSo cells. All buffers can be found in Supplementary Table 2. All steps were carried out at 4 °C or on ice. In brief, $20 \times 10^6$ (total mitochondria) or $80 \times 10^6$ (pure mitochondria devoid of mitochondria-ER contacts) cells were collected by trypsinization and washed once in PBS. Cells were pelleted at $800 \times g$ for 5 min and homogenized in ice-cold BFB1 buffer using an automated Teflon homogenizer. The material was then pelleted twice at $800 \times g$ for 5 min. The cleared supernatant was subjected to $9000 \times g$ for 30 min, the supernatant was used to isolate the endoplasmic reticulum, while the pellet contained the total mitochondrial fraction. When needed, this mitochondrial fraction was collected, mixed again with BFB4 buffer and centrifuged at $6000 \times g$ for 10 min. The pellet was then lysed in RIPA buffer and referred to as "mitochondria". To isolate the ER, the supernatant was centrifuged at $20,000 \times g$ for 20 min and transferred to a fresh tube for an additional centrifugation of $100,000 \times g$ for 1 h. The final ER pellet was lysed in RIPA buffer. To isolate pure mitochondria devoid of the ER-mitochondria contact fraction, the pellet was resuspended gently in BFB2 buffer and centrifuged at $10,000 \times g$ for 10 min. The pellet was then very gently resuspended in BFB3 buffer and centrifuged again at $10,000 \times g$ for 10 min. The pellet was then resuspended in BFB4 buffer and layered on a 30% Percoll gradient (Sigma) and subjected to centrifugation at $95,000 \times g$ for 30 min. The pure mitochondrial fraction was then collected, mixed again with BFB4 buffer and centrifuged at $6000 \times g$ for

10 min. The pellet was then lysed in RIPA buffer and referred to as "pure mitochondria". Proteinase K (Invitrogen) digestions were performed using total mitochondria. Pellets were resuspended in PBS, and equal samples were treated with 1 μg/ml Proteinase K on ice for 30 min. The reaction was stopped using 100 μM PMSF and subjected to centrifugation at $6000 \times g$ for 10 min. The pellet was then lysed in RIPA buffer. For all fractions, protein concentrations were measured by BCA (Invitrogen) and equal protein amount was loaded for the fractions on one SDS-Page gel and subjected for western blotting. Enrichment of the fractions was validated using compartment specific antibodies.

## Oxygen consumption rate

Mitochondrial oxygen consumption rate was quantified using a Seahorse XF96 Analyzer (Agilent) in XF96 cell culture microplates. The evening before the assay, the XF96 sensor cartridge was hydrated in XF calibrant within a $CO_2$-free incubator according to the manufacturer's instructions. On the day of the experiment, cells were exchanged into Seahorse assay medium consisting of unbuffered DMEM supplemented with 25 mM glucose, 1 mM sodium pyruvate, and 3.97 mM L-glutamine (pH 7.2–7.4). OCR was recorded under basal conditions and following sequential injections of oligomycin A (1 μM; Sigma Aldrich), FCCP (1 μM; Sigma Aldrich), and a combined rotenone (Sigma Aldrich) /antimycin A (Sigma Aldrich) treatment (0.5 μM each). Basal respiration, ATP-linked respiration, and maximal respiratory capacity were derived using the Seahorse XF Cell Mito Stress Test analysis in Wave software (v2.6.1, Agilent).

## Proteasome activity

Cells were trypsinized and lysed in a proteasome lysis buffer (25 mM HEPES pH 7.2, 50 mM NaCl, 1 mM $MgCl_2$, 1 mM ATP (Sigma-Aldrich), 1 mM DTT, 10% glycerol, 0.5% Triton X-100) for 30 min at 4 °C and cleared by centrifugation at $20,000 \times g$. Reaction buffer (25 mM HEPES pH 7.2, 50 mM NaCl, 1 mM MgCl2, 1 mM ATP, 1 mM DTT, 10% glycerol) was mixed with 100 μM Suc-Leu-Leu-Val-Tyr-AMC (ENZO Life Sciences). 20 μM MG132 was added to one additional sample as a control for complete inhibition of activity. The 96-well plates were analyzed in a microplate reader FLUOStar OPTIMA (BMG Labtech) at 380 nm/440 nm wavelength every minute for a total of 1 h.

## Cycloheximide experiments

Cells were plated for 24 h and incubated with 20 μg/ml cycloheximide for the indicated times and additionally for 4 h. Cells were then washed once in PBS, trypsinized for 5 min at RT, stopped by 5× the volume of Hanks' Balanced Salt Solution (HBSS with calcium and magnesium), and directly subjected to flow cytometry. The value of the 4-h time point was subtracted to obtain the value of the short-lived population. Every time point was set as a fold-change to $t = 0$ within each cell line.

## SiRNA and plasmid transfections

For transient knockdowns, SilencerSelect siRNAs (Invitrogen) were mixed with RNAiMax (Invitrogen) according to the manufacturer's instructions and seeded together with the cells. The medium was exchanged after 24 h. After 72 h, cells were analyzed. Knockdown efficiencies have been verified by either western blotting or qRT-PCR. For the siRNA mini-screen, three independent SilencerSelect siRNAs (Invitrogen) per gene were used and only selected hit candidates have been verified for knockdown efficiency. For transient plasmid transfections, Lipofectamine 2000 or Lipofectamine 3000 (Invitrogen) has been mixed with plasmids and added to previously plated cells. After 24 h, the medium was changed, and cells were analyzed after a total of 48 h of overexpression or placed in selection medium for stable cell line generation. A list of used siRNAs and plasmids can be found in Supplementary Table 2.

## Quantitative real-time-PCR

Total RNA was extracted using RNeasy mini kit (Qiagen) according to the manufacturer's instructions. Complementary DNA synthesis was carried out using the following protocol: total RNA was mixed with Oligo(dT)$_{18}$ primer and dNTPs (both Invitrogen) and incubated 5 min at 65 °C and 2 min on ice. Then, 5× first strand buffer, 0.1 M DTT and RNAseOUT (all Invitrogen) were added and incubated 2 min at 37 °C. M-MLV-RT was added and incubated for 50 min at 37 °C. The final cDNA was diluted to 10 ng/μl and used for qRT-PCR. Quantitative RT-PCR was performed using TaqMan gene expression master mix (Invitrogen) and assays on demand (Applied Biosystems) with the following primers: Hs00183680_m1 (Rnf139), HS02786624_g1 (Gapdh), Hs01553223_m1 (Rnf121), Hs01060665_g1 (Actin), Hs00744729_s1 (EIF5A) and Hs00702673_s1 (EIF5A2). Cycle threshold (Ct) values were normalized to GAPDH (ΔCt) and relative expression values were calculated using the $2^{(-\Delta Ct)}$.

## Immunofluorescence and imaging

MelJuSo parental and reporter cell lines were grown overnight on coverslips and treated as indicated. Cells were fixed using 4% paraformaldehyde (Invitrogen) for 10 min, permeabilized using 0.2% Triton-X100 in 1× PBS for 20 min, 100 nM glycine was added for 10 min, followed by 3% BSA (Sigma) in 1× PBS for 30 min. The primary antibody was diluted in 0.1% Tween-20 in PBS and incubated overnight at 4 °C. After a 2-h incubation at RT (Invitrogen) in 0.1% Tween-20 in PBS. Nuclear staining was performed using Hoechst 33342 (Molecular Probes) 1:5000 in PBS for 30 min at RT. Fixed cells were examined with the Leica Stellaris 5 confocal laser scanning microscope (63×/1.4 oil objective or 40×/1.25 objective). Image processing was performed with FiJi. Mitochondrial network was traced through Analyze Particles function, and the aspect ratio (ratio between the major and minor axis of mitochondria) was used as an index of mitochondrial length.

## Time-lapse imaging and growth analysis

MelJuSo cell lines were seeded in a 12-well or 6-well plate (Sarstedt) and imaged with the IncuCyte S3 (Sartorius) at constant 37 °C and 5% $CO_2$. Images were taken at time intervals of 2 h, for 24 h. For each well, four different sites were imaged. Cell confluency was quantified with the IncuCyte software (2019B Rev2 version).

## Aggregates imaging and quantification

Cells were transfected with the indicated plasmids, after 6 h detached from the plate and re-seeded on coverslips. 48 h after transfection, cells were fixed using 4% paraformaldehyde (Invitrogen) for 10 min, and nuclei were stained using Hoechst 33342 (Molecular Probes) 1:5000 in PBS for 30 min at RT. Fixed cells were examined with a Leica Stellaris 5 confocal laser scanning microscope (20x/0.75 air objective). Four images were taken and cells with aggregates were counted per expression construct and per experiment and related to the total number of nuclei per image.

## Statistics

Statistical analyzes were performed using Prism GraphPad v10. The number of experiments is indicated in the Figure legends and present biological independent experiments. Error bars present ±SD (standard deviation). Statistical tests are indicated for each experiment in the Figure legends and in the Source Data files. Exact $P$-values are presented in the Source Data files. The following $P$-values were considered significant: $^{*}P \leq 0.05$, $^{**}P \leq 0.01$, $^{***}P \leq 0.001$, $^{****}P \leq 0.0001$.

## Reporting summary

Further information on research design is available in the Nature Portfolio Reporting Summary linked to this article.

## Data availability

All data supporting the results of this study can be found in the article, supplementary, and source data files. Source Data are provided with this paper. The sequences from CRISPR-Cas9 knockout screen are deposited in the Gene Expression Omnibus (GEO) under accession number GSE328470. Source data are provided with this paper.

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

## Acknowledgements

We thank Maria Masucci and the members of the Dantuma lab for helpful input and the Biomedicum Imaging Core Facility and the Biomedicum Flow Cytometry Core Facility for their assistance. Part of this work was performed at the CRISPR Functional Genomics unit, funded by SciLifeLab and Karolinska Institutet. We thank Lukas Mann for help with cloning the tet-off plasmid. This work was supported by the Swedish Research Council (N.P.D. 2021-02562, L.N. 2023-02503), the Swedish Cancer Society (N.P.D. CAN 211653Pj), the Swedish Brain Foundation (FO2022-0271, F2023-0376), Leif Lundblad Family private donation (L.N.) and the Karolinska Institute (KID grant). M.E.G. was supported by research

fellowships from the Deutsche Forschungsgemeinschaft (DFG) (GI-1329/1-1). N.P.D. is a member of the COST network ProteoCure.

## Author contributions

M.E.G., N.P.D. conceptualized the study; M.E.G, E.B., M.M., M.H.W. performed the experiments; L.N. assisted in the Seahorse and fractionation experiment; F.A.S. assisted in microscopy and image analysis; M.E.G, E.B., M.M., M.H.W., M.A., L.N., F.A.S., N.P.D. analyzed the data; M.E.G., N.P.D. wrote the manuscript draft; M.E.G., N.P.D. coordinated the project; All authors edited and approved the final manuscript.

## Funding

## Competing interests

The authors declare no competing interests.
