## [Transparent Peer Review file · Nature Communications]

Mitochondria serve as a Holdout Compartment for Aggregation-Prone Proteins hindering Efficient Degradation

Corresponding Author: Professor Nico Dantuma

Version 0:

Reviewer comments:

Reviewer #1

(Remarks to the Author)

The authors have performed a genome-wide CRISPR/Cas9 KO screen to identify specific regulators of the UPS dependent degradation of aggregation prone reporter protein YFP-CL1, providing a valuable dataset. The authors identified an interesting link between eIF5A perturbations and proteostasis, however the manuscript requires additional experimental revisions to strengthen the claims and provide more mechanistic details.

1. In eukaryotes, eIF5A appears to have two isoforms that share a very high amino acid sequence identity. However, they are expressed under different conditions, have distinct cellular localisation and function. The authors should discuss this in the manuscript. Given the high sequence similarity, have the authors validated whether the eIF5A siRNA targets both isoforms and also whether the sgRNAs also targeted both isoforms in the CRISPR screen and in the eIF5A KO cell line.

2. The authors used YFP-CL1 reporter, which is a short-lived UPS reporter (as confirmed by CHX chase experiments in Fig 1B), however the CRISPR/Cas9 KO screen ran for 10-18 days depending on the replicate. The authors should discuss that such a long-time frame could identify factors that can indirectly modulate long-lived aggregated forms of the protein and/or adaptive cellular stress responses. This links to the comment below (3.) - when does autophagy become engaged in the process – will this be at later stages? The state of aggregation of the YFP-CL1 reporter is missing. Can the authors clarify whether the reporter remains soluble or becomes aggregated over time (10-18 days) under the studied conditions.

3. Determining the contribution of the UPS versus autophagy is important. Based on Bafilomycin A1 treatment the authors conclude that autophagy does not contribute to YFP-CL1 clearance. Although the concentration and treatment time are relatively standard to block lysosomal acidification, no data are included to show evidence of autophagy flux inhibition in the studied system. As a minimum authors should measure autophagic flux: LC3II and p62 levels involving use of BafA1 and chloroquine.

4. The claim that YFP-CL1 localizes to mitochondria is key to the proposed model, but this needs to be further validated. Is YFP-CL1 in close proximity to mitochondria or imported into mitochondria? The authors described the YFP-CL1 reporter being distributed throughout the cytosol and nucleus, and I believe this reporter was used in Fig. 6A-B for ER and mitochondrial fractionation. The use of compartment specific reporters (i.e. mitochondrial targeting sequence-CL1) and specific mitochondrial fractionation will strengthen the claims.

5. eIF5A has a role in CAT-tailing and ribosome quality control (RQC) of nuclear-encoded mitochondrial proteins (PMID: 29107329). Loss of eIF5A in yeast leads to increased ribosomal stalling and subsequent accumulation of import stalled mitochondrially destined polypeptides at the OMM. Could this pathway be an important regulator in the proposed model? Does the YFP-CL1 signal localises to OMM and not the other mitochondrial compartments? The above proposed mitochondrial subfractionation should be performed.

Minor revisions:

P1 line 1-2:

Title – further experimental studies are required to confidently state that ‘Mitochondria serve as a Holdout Compartment for Aggregation-Prone Proteins hindering Efficient Degradation’

P3 line 48:
(ref 3) please add more up to date references.

P3 line 60-65:
Authors should acknowledge the role of autophagy in the process of removing protein aggregates in Huntington's disease and spinocerebellar ataxia type 7, where the larger protein aggregates are primarily removed by autophagy, and the UPS will handle smaller, more soluble and short-lived proteins.

P4 line 96 and P8 line 225:
I would tone down the following statement to perhaps 'modulate' : 'eIF5A, a master regulator of mitochondrial homeostasis'. When refereeing to eIF5A as 'general regulator of mitochondrial homeostasis (42)' - there is a need for further studies to delineate the exact mechanism(s) through which eIF5A influences mitochondrial homeostasis and perhaps cite specific studies rather than a review (ref 42) – which would be much more suitable.

P6 lines 141-150 and P18 methods section:
Please provide more details or comment on the genome-wide CRISPR/Cas9 KO screen (i.e. fold library coverage, statistical cut off for the selection of top hits, FDRs). Why were the cells for FACS sorting cultured for different number of days – these are not true replicates. Was replicate 1 a pilot screen and replicate two the actual discovery screen?

Why in Fig 6A-B different mitochondrial markers were used i.e. TIM50 and TOM20?

Figures 6C-D and Suppl. Figure 6A:
Please provide the number of cells analysed in Figures 6C-D and Suppl. Figure 6A. Is there a quantification for the images in Suppl. Figure 6A? eIF5A KO cells appear to have a very dense mitochondrial network on the Fiji masking – was this quantifiable? Does eIF5A KO or GC7 treatment cause early loss of mitochondrial membrane potential (within hours) or is the membrane potential intact and the mitochondrial fragmentation appears at later stages, which would suggest that is a secondary effect?

The discussion could be expanded for example is there a correlation between eIF5A and TIM50 and this study; and eIF5A also facilitates translation elongation of Atg7.

P21 line mitochondrial fractionations – this also includes ER fractionation please reflect in the heading and refer to the supplementary table 3 for the BFB1-4 buffers.

Supplementary Table 3 – please clarify what are the dilutions on p4 and s22179 etc. identifiers. There are major formatting errors from P4 onwards

Reviewer #2

(Remarks to the Author)
Manuscript: Nature Communications manuscript NCOMMS-25-56601

Title: Mitochondria serve as a Holdout Compartment for 1 Aggregation-Prone Proteins hindering Efficient Degradation
Authors: Maria E. Gierisch^{1,*}, Enrica Barchi¹, Mirco Marogna¹, Moritz H. Wallnöfer¹, Maria Ankarcrona², Luana Naia², Florian A. Salomons¹ and Nico P. Dantuma^{1,*}

Reviewed by Paula Alepuz and co-reviewed by the early career investigator Marina Barba-Aliaga

The work by Gierisch et al. aims to identify regulators of the degradation of aggregation-prone proteins, with the ultimate goal of enhancing degradation systems as a therapeutic approach to eliminate proteins whose aggregation causes neurodegenerative diseases. The authors set up an interesting model of an aggregation-prone reporter protein, in which yellow fluorescent protein (YFP) is fused to an amphipathic helix that makes the protein prone to aggregation (YFP-CL1). The same mRNA that contains YFP-CL1 also co-expresses a histone fused to BFP. This system allows yellow and blue fluorescence to be normalised, so it is expected to provide information about protein degradation. Using this reporter, the authors performed a genome-wide CRISPR-Cas9 screening in mammalian cells followed by flow cytometry analysis. They were able to identify mutants with higher and lower levels of YFP-CL1. Firstly, they identified mutants in the UPS pathway and ER-associated ubiquitin ligases, the mutation of which produces higher YFP levels. This led them to conclude that the ER-cytosolic membrane is implicated in the degradation of YFP-CL1, and also validated the screening. Secondly, the authors identified genes whose mutations yielded lower YFP levels. Among those stands the translation elongation factor eIF5A. The authors then present experiments that suggest that, under normal conditions, part of the YFP-CL1 protein is linked to mitochondria, which prevents its degradation by the UPS pathway. This mitochondrial attachment depends on the amphipathic helix and the function of eIF5A. In general, the manuscript is clearly structured, and easy to follow. The authors present their findings in a logical and coherent manner, with a clear message that is accessible to a broad readership.

Overall, we think this study is interesting and establishes a robust screening system. However, the role of mitochondria as a compartment for proteins prone to aggregation, and the role of eIF5A in keeping these proteins attached to the mitochondria,

need to be supported by more experiments to justify the strong statements in the title and abstract. The main four issues to resolve are: 1) Using microscopy, clearly demonstrate the general role of mitochondria as a holdout compartment for aggregating proteins; 2) Rule out the possibility of indirect mitochondrial effects on proteasome activity due to lower cellular ATP levels, increase aggregation of mitochondrial proteins in the cytoplasm, or other reasons; 3) Rule out the possibility of an eIF5A role in synthesising this aggregating protein due to an amphipathic helix, and propose a mechanism by which eIF5A links this aggregating protein to mitochondria; 4) The results showing the effect of eIF5A on htt mutant protein levels based on the presence of an amphipathic helix that keeps htt associated with mitochondria should be strengthened.

Next, we will explain the main issues in more detail and suggest some specific experiments, We also include minor comments/suggestions.

Major issues:

1. In order to strongly support the statement that 'mitochondria serve as a holdout compartment for aggregation-prone proteins', more direct evidence based on protein localisation experiments should be presented.

1.1 First, the localisation of YFP-CL1 in mitochondria should be investigated using microscopy and specific mitochondrial markers such as MitoTracker and Tom20-GFP. This should be done under normal conditions, where it is shown that YFP-CL1 is distributed in the cytosol, but it is expected that around 20% of the protein would be linked to the mitochondria, as suggested by the authors' results.

1.2 Suppl. Figure 1B: Treatment with expomicin/TAK-243 results in increased YFP-CL1 fluorescence and the formation of perinuclear foci. However, it is surprising that the authors do not present any experiments demonstrating the colocalisation of this fluorescent signal with mitochondria, which they propose as a cellular compartment for holding these aggregates. Including such colocalisation data would strengthen their argument.

1.3 The localisation of the YFP-CL1 construct to mitochondria should also be investigated under conditions of eIF5A inhibition and mitochondrial dysfunction (OXPHOS inhibitors). Visualising mitochondria with a protein-independent marker such as Mitotracker (in addition to Tom20 localisation) is of interest, given that mitochondrial protein mislocalisation and aggregation in the cytoplasm have been reported under eIF5A depletion.

1.4 Controls using other reporters fused to different amphipathic helices should be employed to confirm the general role of mitochondrial attachment and eIF5A function in preventing the degradation of these aggregation-prone targets. One interesting reporter would be an endogenous cytosolic protein fused to CL1 and bearing an HA or myc epitope, as well as YFP fused to the htt amphipathic helix. This would allow protein levels to be determined, with direct or indirect immunofluorescent microscopy being used to study the localisation of the fused proteins. Alternatively, subcellular fractionation could be employed to investigate the involvement of mitochondrial attachment and eIF5A in the regulation of protein stability.

1.5 With respect to the subcellular compartment fractionating shown in Fig.6, to conclude "YFP-CL1 was readily detected in both ER and mitochondrial fractions, indicating that the amphipathic helical CL1 interacts with both membranous compartments (Fig. 6A)", it should be shown a control of YFP localization with the helix modified version, YFP-CL1*. In Fig. 6A, why is there less Tim50 in mitochondrial fraction when expression YFP-CL1??? Show Ponceau in Fig.6A and also use the control Tim20. Additionally, it should be quantified the protein levels in Fig. 6A and B, with respect to controls, from several experiments.

In relation with this, in Fig. 5D and E it is shown NES-GFP-CL1 and NLS-GFP-CL1 reporters (the first affected and the second not by eIF5A), but in the text says: "we found that cytosolic localization of the reporter was a prerequisite for enhanced degradation induced by GC7 treatment (Fig. 5D), as the nuclear YFP-CL1 reporter was unresponsive (Fig. 5E)". So, where is the mistake? Are GFP or YFP constructs used? This should be clarified and the experiment proposed above using different reporters should be done.

2. Additional experiments should be performed to exclude the possibility of an indirect effect of eIF5A mediated by a drop in cellular ATP levels caused by loss of mitochondrial activity under eIF5A depletion, and to discard a direct or indirect role of eIF5A in proteasome activity.

2.1 Experiments with ATP depletion in native cells and under low active eIF5A conditions should be done to inform of ATP dependent/independent role of eIF5A. It could be investigated the localization/stability of the reporter YFP-CL1.

2.2 Measure proteasomal activity with specific proteasomal degradation reporters in native cells and under eIF5A depletion, also with inhibitors of mitochondrial activity. This will allow to directly exclude an effect of eIF5A in proteasomal activity.

2.3 eIF5A is supposed to be essential in all eukaryotic cells, so the isolation of mutants by CRISPR-Cas9 screening and also the subsequent isolation of eIF5A knockdown clones could carriage additional suppressor mutations. Therefore, one should be cautious and check most results with other ways of eIF5A inhibition. Experiments of loss of mitochondrial localization of YFP-CL1 (Fig.6B) should be also confirmed with depletion of eIF5A protein by using siRNA against eIF5A and/or GC7 treated cells.

3. The eIF5A main role described is as translation elongation factor, therefore a direct role in the synthesis of YFP-CL1 must be ruled out.

3.1 In their system, the authors co-express YFP-CL1 and H2A-BFP in the same mRNA. Therefore, normalising the YFP signal against the BFP signal eliminates the effects of transcription. However, it cannot be assumed that the effects of translation/ protein synthesis are not affecting the results, since translation rates may differ between the two open reading frames (ORFs) under active eIF5A depletion. The authors demonstrate that blocking the proteasomal degradation of YFP-CL1 appears to reverse the effects of eIF5A depletion. However, a more direct test is required to rule out the possibility of an eIF5A role in YFP-CL1 synthesis. To demonstrate that the effect of eIF5A is linked to stability rather than synthesis, it would be necessary to directly analyse YFP-CL1 protein stability in eIF5A-depleted cells (using siRNA against eIF5A and GC7 treatment) versus native conditions, employing a conventional translation shut-down experiment involving CHX and the detection of protein levels over time. This experiment should also be performed with the YFP-CL1* version with a modified amphipathic helix to control for any differences in levels and stability.

3.2 From Fig. 3F it is stated that “depletion of eIF5A selectively reduced the steady-state levels of YFP-CL1 without a substantial decrease in the levels of ERAD and soluble substrates (Fig. 3F)”. However, if eIF5A plays a role in the translation elongation of the reporter, this may depend on the position of YFP within the open reading frame (ORF). On the one hand, it could be checked whether CL1-YFP level is affected by eIF5A inhibition; on the other hand, at least some of the ERAD and soluble reporters should be examined with YFP in the N-terminal part of the protein.

4. The results with mutant htt are somewhat weak. In Figures 8D and 8E, the effect of proteasome inhibition on suppressing the effect of eIF5A is minimal. A similarly weak difference is reported when the amphipathic helix of htt is mutated.

4.1 We suggest probing the role of eIF5A in linking htt to mitochondria and regulating its stability through this mitochondrial attachment more directly. Biochemical fractionation and/or microscopy experiments should be conducted to determine whether htt is partially linked to mitochondria in an eIF5A-dependent manner, and whether this link depends on the htt amphipathic helix. The experiments should be quantified.

5. No molecular mechanism is suggested for the role of eIF5A in the mitochondria. Do the authors think that eIF5A facilitates the retention of the aggregation-prone peptides in the mitochondria? Is it helping the import of these peptides through the mitochondrial membranes?

5.1 To begin addressing this, co-immunoprecipitation experiments of eIF5A using mitochondrial fractions could provide insights. Additionally, more discussion of the potential eIF5A-mediated molecular mechanism would strengthen the manuscript and should be included in the Discussion section.

5.2 It has been recently shown that eIF5A depletion causes mitochondrial import defects because the stall of ribosomes in TIM50 mRNA, which encodes a protein of the MIM system. This defect in mito-protein import to the mitochondria could cause the attachment of prone-aggregation proteins to the mitochondrial surface?? To address this, it could be examined the localization/stability/levels of YFP-CL1 and YFP-CL1* and/or mutant htt under conditions of depletion of Tim50 protein.

Other Major concerns:

6. The authors do not mention anywhere that eIF5A is encoded by two isoforms (EIF5A1 and EIF5A2) neither which isoform is targeted by the three silencing RNAs or deleted in the KO cell line. This distinction is important and should be clearly stated in the main text and figure legends throughout the manuscript.

7. In this line, the authors do not mention anywhere which is the most relevant function of eIF5A in translation (relieving ribosome stalling at specific peptide motifs). This should be acknowledged in the main text.

8. Several experiments were performed using three individual silencing RNAs targeting eIF5A, yet none of them triggered a strong reduction in YFP-CL1 levels. Would it be possible to combine the three siRNAs simultaneously to better increase the knockdown efficiency and drive more pronounced effects on protein degradation? Has this been already tested? If this approach has already been tested, it should be mentioned and discussed.

9. Cell viability should be assessed following 24-hour treatments with different concentrations of GC7 to ensure that, although cell proliferation might be potentially reduced, the cells remain viable. This control is important to confirm that the observed effects are not simply due to cytotoxicity.

Minor issues/suggestions:

1. Keywords suggested: mitochondria, eIF5A

3. Figure 1B. Band quantification, replicates, error bars and statistics are missing.

4. Figure 2B and 3B. The software or tool used to achieve the GO representation should be better included in the figure legend.

5. Figure 2D. Please include the concentration used for the siRNAs in the figure legend.

6. Figures 2E,F. Statistics relative to control as well as individual dots for each replicate value are missing.
8. Figure 5A. Statistics are missing.
9. Figures 5B,C. Statistical comparisons between untreated and GC7 treatment are missing.
10. Figure 5G. Change to a classic bar graph with individual dots for each replicate value to maintain the general graphic style of the paper.
12. Log fold change: sometimes written as "lfc" and other times as "LFC". Please unify.
13. In general, most of the figures show the signal for YFP fluorescence measured by flow cytometry, but it is not specified if this is calculated by normalizing this signal to the tBFP one. Please include this information in every figure legend.
14. In general, there are many experiments with a lot of replicate values for the control (more than 5 most of the cases) while 2 or 3 values for the rest of the conditions. At least, 3 independent values should be shown for each experiment and better use same number of replicas.
15. Line 255: "eIF5A depletion significantly reduced YFP-CL1 levels..." Include which is the percentage of signal decrease.
16. Line 311, reference 42. Add the following references where it is demonstrated the eIF5A induces a metabolic switch from oxidative phosphorylation to glycolysis (<https://doi.org/10.1002/1873-3468.14366> and <https://doi.org/10.3390/ijms22010219>).
17. Line 329. "And the mitochondrial compartment interfering with this process." Suggestion to add that this is also mediated by the action of eIF5A.
18. Line 389: "N-mut htt-GFP inclusions measured by flow cytometry". Explain a little bit more in detail about how this is performed.
19. Please, add some information about Hoescht staining in the figure legends.
20. Figure 7B. Legend only specifies anti-GFP antibody, is anti-GFP or YFP?? Please, add details corresponding to other antibodies used here.
21. Figure 7D. Add control images with no proteasome inhibition.
22. Figure 8A. Legends says n=3 but clearly is higher.
23. Figure 8E. Statistical comparisons between untreated and GC7 mut htt are missing.
24. Figure 9. Please include eIF5A protein in the model representation since it is already in the figure legend.
25. Materials and Methods, Western Blotting section. Add conditions of centrifugation steps. Include how protein quantification is performed as well as the amount of proteins loaded on SDS-PAGE.
26. Supplementary Table 3. "Reciepe" change to recipe.
27. As mentioned above in Fig.5D, F (NES-GFP)-CL1, (NLS-GFP)-CL1, It should be clarify whether GFP or YFP has been use and correct text and figure accordingly.
28. Fig5F figure legend "plotted for their tBFP versus YFP ratio", should it be the opposite?
29. Fig. 6E Y-axis is written "aspect ratio", it should be better described what is measured: length of the mitochondrial filaments.
30. Supplementary Figure 6E, include control of cells with eIF5A depletion.

Reviewer #3

(Remarks to the Author)

Version 1:

Reviewer comments:

Reviewer #1

(Remarks to the Author)

The authors have done an excellent job addressing all my concerns with substantial new experiments, which significantly strengthens the manuscript. I have only two minor points that should be addressed before final acceptance:

1. Proteinase K interpretation (Suppl. Fig. 7c, lines 365-368): The interpretation might be slightly overreaching. Could the authors either add other control blots or soften the language in lines 365-368 to acknowledge the protein could also reside in the intermembrane space if the outer membrane was compromised during isolation.
2. Please check the reference update (line 48, ref 3): The authors stated in their rebuttal that "This citation has been replaced with a review from 2024," but the current reference in the manuscript remains: "Soto, C. Unfolding the role of protein misfolding in neurodegenerative diseases. *Nat Rev Neurosci* 4, 49-60, (2003)." Please update this citation.

Reviewer #2

(Remarks to the Author)

Manuscript: Nature Communications manuscript NCOMMS-25-56601A

Title: Mitochondria serve as a Holdout Compartment for 1 Aggregation-Prone Proteins hindering Efficient Degradation

Authors: Gierisch et al.

Reviewed by Paula Alepuz and co-reviewed by the early career investigator Marina Barba-Aliaga

The authors have convincingly replied to most of the raised concerns and also presented a large number of new experiments to support their statements. We only have one minor concern regarding concern 4.1, which arose in our previous revision. The author did not answer the question of whether N-Htt protein association with mitochondria is eIF5A-dependent, as they have shown for YFP-CL1. A similar experiment to the one shown in Fig. 6c would be required.

Marina Barba-Aliaga co-reviewed this manuscript with one of the reviewers who provided the listed reports. This is part of the Nature Communications initiative to facilitate training in peer review and to provide appropriate recognition for Early Career Researchers who co-review manuscripts.

Reviewer #3

(Remarks to the Author)

Version 2:

Reviewer comments:

Reviewer #1

(Remarks to the Author)

The authors have satisfactorily addressed all outstanding concerns that were raised in the previous revision.

Reviewer #2

(Remarks to the Author)

The authors have convincingly replied to our last concern and the new experiments supports their statement.

Reviewer #3

(Remarks to the Author)

Rebuttal NCOMMS-25-56601 “Mitochondria serve as a Holdout Compartment for Aggregation-Prone Proteins hindering Efficient Degradation”

Reviewer #1: The authors have performed a genome-wide CRISPR/Cas9 KO screen to identify specific regulators of the UPS dependent degradation of aggregation prone reporter protein YFP-CL1, providing a valuable dataset. The authors identified an interesting link between eIF5A perturbations and proteostasis, however the manuscript requires additional experimental revisions to strengthen the claims and provide more mechanistic details.

Authors: We thank the reviewer for the insightful comments. Additional experiments have been performed and the text has been edited to address the comments of the reviewer.

1. In eukaryotes, eIF5A appears to have two isoforms that share a very high amino acid sequence identity. However, they are expressed under different conditions, have distinct cellular localisation and function. The authors should discuss this in the manuscript. Given the high sequence similarity, have the authors validated whether the eIF5A siRNA targets both isoforms and also whether the sgRNAs also targeted both isoforms in the CRISPR screen and in the eIF5A KO cell line.

Authors: The EIF5A gene encodes two splice variants that give rise to two different variants of the eIF5A protein (Gene: EIF5A2 (ENSG00000163577) - Summary - Homo sapiens - Ensembl genome browser 115). Moreover, there is a tissue-specific paralog EIF5A2 that we assume the reviewer is referring to (Gene: EIF5A2 (ENSG00000163577) - Summary - Homo sapiens - Ensembl genome browser 115). Our siRNAs target both splice variants encoded by EIF5A but do not have sequence specificity for the EIF5A2 paralog. We have performed quantitative PCR that shows that the siRNAs selectively knockdown eIF5A transcripts with little or no effect on the marginal levels of eIF5A2 transcripts in MeJuSo cells. There was no significant increase in eIF5A2 in one EIF5A knockout cell line and a minor but significant increase in the marginal levels of eIF5A2 in the second EIF5A knockout cell line. As both eIF5A and eIF5A2 are dependent on hypusination, the absence of a hypusine signal in cells transfected with eIF5A-targeting siRNAs and EIF5A knockout cells confirms that overall eIF5A activity is ablated in these cells.

The results of the quantitative PCR are shown in Suppl. Fig. 4a-c and Suppl. Fig. 6b. The results are described in lines 261-267 and 321-325.

2. The authors used YFP-CL1 reporter, which is a short-lived UPS reporter (as confirmed by CHX chase experiments in Fig 1B), however the CRISPR/Cas9 KO screen ran for 10-18 days depending on the replicate. The authors should discuss that such a long-time frame could identify factors that can indirectly modulate long-lived aggregated forms of the protein and/or adaptive cellular stress responses. This links to the comment below (3.) - when does autophagy become engaged in the process – will this be at later stages? The state of aggregation of the YFP-CL1 reporter is missing. Can the authors clarify whether the reporter remains soluble or becomes aggregated over time (10-18 days) under the studied conditions.

Authors: The YFP-CL1 reporter is commonly used as a reporter UPS substrate. It gives a soluble distribution throughout the cytoplasm and nucleus and does not form detectable aggregates in reporter cells unless the ubiquitin-proteasome system is inhibited or proteotoxic stress is induced. Even after culturing the cells for 10-18 days (or longer), there are no detectable aggregates in the cells or signs of toxicity due to YFP-CL1 aggregation.

We clarify this in lines 121-125.

3. Determining the contribution of the UPS versus autophagy is important. Based on Bafilomycin A1 treatment the authors conclude that autophagy does not contribute to YFP-CL1 clearance. Although the concentration and treatment time are relatively standard to block lysosomal acidification, no data are included to show evidence of autophagy flux inhibition in the studied system. As a minimum authors should measure autophagic flux: LC3II and p62 levels involving use of BafA1 and chloroquine.

Authors: We have included western blots of LC3II and p62 that confirm that autophagy is inhibited under the conditions that we used. We also confirm in the same samples that inhibition of autophagy, unlike UPS inhibition, does not cause a significant increase in YFP-CL1 levels.

See Suppl. Fig. 1b and 5b and line 293.

4. The claim that YFP-CL1 localizes to mitochondria is key to the proposed model, but this needs to be further validated. Is YFP-CL1 in close proximity to mitochondria or imported into mitochondria? The authors described the YFP-CL1 reporter being distributed throughout the cytosol and nucleus, and I believe this reporter was used in Fig. 6A-B for ER and mitochondrial fractionation. The use of compartment specific reporters (i.e. mitochondrial targeting sequence-CL1) and specific mitochondrial fractionation will strengthen the claims.

Authors: Mitochondrial fractionation confirmed that pure mitochondria fractions (devoid of ER or ER-mitochondria contact sites) contain YFP-CL1. In addition, we performed microscopic analysis and show that YFP-CL1, but not the control reporter Ub-YFP, co-localizes with a mitochondrial marker and MitoTracker.

See Fig. 6a and Suppl. Fig 7a and text 347-350.

We aimed to engineer a compartment specific reporter through introduction of an N-terminal mitochondrial localization signal in the YFP-CL1. Unfortunately, this fusion was expressed at very low levels and could not be used for addressing this question. Serendipitously, we found that N-terminal positioning of the CL1 amphipathic helix (CL1-YFP) resulted in efficient localization of the fusion to mitochondria. This coincided with reduced proteasomal degradation, in line with our hypothesis.

See Suppl. Fig. 7d,e and lines 368-372.

The reviewer brings up an interesting point with the possibility that YFP-CL1 may be imported into mitochondria. We have performed proteinase K treatment of the mitochondria fraction, which shows that the mitochondria-associated YFP-CL1 is easily accessible, arguing against mitochondrial import of the reporter.

See Suppl. Fig. 7c and lines 364-368.

5. eIF5A has a role in CAT-tailing and ribosome quality control (RQC) of nuclear-encoded mitochondrial proteins (PMID: 29107329). Loss of eIF5A in yeast leads to increased ribosomal stalling and subsequent accumulation of import stalled mitochondrially destined polypeptides at the OMM. Could this pathway be an important regulator in the proposed model? Does the YFP-CL1 signal localises to OMM and not the other mitochondrial compartments? The above proposed mitochondrial subfractionation should be performed.

Authors: This is indeed a very interesting point that we have been considering. However, depletion of LTN1 and ZNF598, two ubiquitin ligases involved in RQC, did not prevent the enhanced degradation of YFP-CL1 in GC7-treated cells. This supports our model that the enhanced degradation of YFP-CL1 is facilitated by its cognate ubiquitin ligases at the ER.

See Suppl. Fig. 5c,d and text 297-301.

Minor revisions:

P1 line 1-2:

Title – further experimental studies are required to confidently state that ‘Mitochondria serve as a Holdout Compartment for Aggregation-Prone Proteins hindering Efficient Degradation’

Authors: Additional experimentation on mitochondrial localization and the role of ubiquitin ligases are included. See above.

P3 line 48:

(ref 3) please add more up to date references.

Authors: This citation has been replaced with a review from 2024.

P3 line 60-65:

Authors should acknowledge the role of autophagy in the process of removing protein aggregates in Huntington's disease and spinocerebellar ataxia type 7, where the larger protein aggregates are primarily removed by autophagy, and the UPS will handle smaller, more soluble and short-lived proteins.

Authors: This is indeed a very relevant point. We mention this now in the introduction and also discuss in more detail the link between eIF5A and autophagy in the discussion.

See lines 57-60 and lines 555-558.

P4 line 96 and P8 line 225:

I would tone down the following statement to perhaps ‘modulate’ : ‘eIF5A, a master regulator of mitochondrial homeostasis’. When refereeing to eIF5A as ‘general regulator of mitochondrial homeostasis (42)’ - there is a need for further studies to delineate the exact mechanism(s) through which eIF5A influences mitochondrial homeostasis and perhaps cite specific studies rather than a review (ref 42) – which would be much more suitable.

Authors: We agree that the regulatory role of eIF5A is not fully understood. As proposed by the reviewer, we are now referring to eIF5A as a modulator of mitochondrial homeostasis.

P6 lines 141-150 and P18 methods section:

Please provide more details or comment on the genome-wide CRISPR/Cas9 KO screen (i.e. fold library coverage, statistical cut off for the selection of top hits, FDRs). Why were the cells for FACS sorting cultured for different number of days – these are not true replicates. Was replicate 1 a pilot screen and replicate two the actual discovery screen?

Authors: The requested details are provided in the Methods section. The days for sorting are indeed not identical but decided based on practical issues (required cell number, availability of flow cytometry sorter at our core facility). In the revision, we are now referring to these as “experiment 1” and “experiment 2” instead of replicates.

Why in Fig 6A-B different mitochondrial markers were used i.e. TIM50 and TOM20?

Authors: The choice of mitochondrial marker was based on practical issues such as its molecular weight and availability of fluorescent antibodies that allowed simultaneous detection. Both proteins are well established as markers for mitochondria.

Figures 6C-D and Suppl. Figure 6A:

Please provide the number of cells analysed in Figures 6C-D and Suppl. Figure 6A. Is there a quantification for the images in Suppl. Figure 6A? eIF5A KO cells appear to have a very dense mitochondrial network on the Fiji masking – was this quantifiable? Does eIF5A KO or GC7 treatment cause early loss of mitochondrial membrane potential (within hours) or is the membrane potential intact and the mitochondrial fragmentation appears at later stages, which would suggest that is a secondary effect?

Authors: The effect of GC7 treatment or eIF5A deletion is striking but unfortunately not easy to reliably quantify when using a 2D plane by confocal microscopy and no high-resolution imaging. We have quantified this and while a similar tendency is observed for GC7-treated cells and EIF5A knock-out cells, the difference is only statistically significant for the former.

See Suppl. Fig. 8a,b and Suppl. Fig. 9b. and lines 378-381.

Regarding the effect of GC7 and eIF5A knockout on membrane potential, we have now included Seahorse experiments to assess the integrity of mitochondria. GC7 results in a strong reduction of oxidative phosphorylation.

See Suppl. Fig. 8c,d and text 381-385.

The discussion could be expanded for example is there a correlation between eIF5A and TIM50 and this study; and eIF5A also facilitates translation elongation of Atg7.

Authors: We extended the discussion on TIM50 and mention that human TIM50, unlike yeast TIM50, does not contain a proline stretch or any other motif that would suggest that its translation is dependent on eIF5A.

See lines 546-550.

We also mention now the role of eIF5A in translation of TFEB, which is a regulator of autophagy. It has been shown that the effect of eIF5A through TFEB translation is dependent of Atg7. We are, however, not aware of publications that show that translation of Atg7 is stimulated by eIF5A.

See lines 555-558.

P21 line mitochondrial fractionations – this also includes ER fractionation please reflect in the heading and refer to the supplementary table 3 for the BFB1-4 buffers.

Authors: ER fractionation has been included. We state that all buffers can be found in Suppl. Table 3.

Supplementary Table 3 – please clarify what are the dilutions on p4 and s22179 etc. identifiers. There are major formatting errors from P4 onwards

Authors: Dilutions have been clarified as well as the identifier numbers for the siRNAs. We are not sure what the formatting errors are that the reviewer is referring to. We can provide the Excel file in a different format if needed.

Reviewer #2

Reviewed by Paula Alepuz and co-reviewed by the early career investigator Marina Barba-Aliaga

The work by Gierisch et al. aims to identify regulators of the degradation of aggregation-prone proteins, with the ultimate goal of enhancing degradation systems as a therapeutic approach to eliminate proteins whose aggregation causes neurodegenerative diseases. The authors set up an interesting model of an aggregation-prone reporter protein, in which yellow fluorescent protein (YFP) is fused to an amphipathic helix that makes the protein prone to aggregation (YFP-CL1). The same mRNA that contains YFP-CL1 also co-expresses a histone fused to BFP. This system allows yellow and blue fluorescence to be normalised, so it is expected to provide information about protein degradation. Using this reporter, the authors performed a genome-wide CRISPR-Cas9 screening in mammalian cells followed by flow cytometry analysis. They were able to identify mutants with higher and lower levels of YFP-CL1. Firstly, they identified mutants in the UPS pathway and ER-associated ubiquitin ligases, the mutation of which produces higher YFP levels. This led them to conclude that the ER-cytosolic membrane is implicated in the degradation of YFP-CL1, and also validated the screening. Secondly, the authors identified genes whose mutations yielded lower YFP levels. Among those stands the translation elongation factor eIF5A. The authors then present experiments that suggest that, under normal conditions, part of the YFP-CL1 protein is linked to mitochondria, which prevents its degradation by the UPS pathway. This mitochondrial attachment depends on the amphipathic helix and the function of eIF5A. In general, the manuscript is clearly structured, and easy to follow. The authors present their findings in a logical and coherent manner, with a clear message that is accessible to a broad readership.

Overall, we think this study is interesting and establishes a robust screening system. However, the role of mitochondria as a compartment for proteins prone to aggregation, and the role of eIF5A in keeping these proteins attached to the mitochondria, need to be supported by more experiments to justify the strong statements in the title and abstract. The main four issues to resolve are: 1) Using microscopy, clearly demonstrate the general role of mitochondria as a holdout compartment for aggregating proteins; 2) Rule out the possibility of indirect mitochondrial effects on proteasome activity due to lower cellular ATP levels, increase aggregation of mitochondrial proteins in the cytoplasm, or other reasons; 3) Rule out the possibility of an eIF5A role in synthesising this aggregating protein due to an amphipathic helix, and propose a mechanism by which eIF5A links this aggregating protein to mitochondria; 4) The results showing the effect of eIF5A on htt mutant protein levels based on the presence of an amphipathic helix that keeps htt associated with mitochondria should be strengthened.

Next, we will explain the main issues in more detail and suggest some specific experiments, We also include minor comments/suggestions.

Authors: We thank Paula Alepuz and Marina Barba-Aliaga for their helpful comments. A considerable amount of additional experimentation has been performed to address the concerns. We feel that the new data strengthen our conclusion and has improved the study.

Major issues:

1. In order to strongly support the statement that 'mitochondria serve as a holdout compartment for aggregation-prone proteins', more direct evidence based on protein localisation experiments should be presented.

1.1 First, the localisation of YFP-CL1 in mitochondria should be investigated using microscopy and specific mitochondrial markers such as MitoTracker and Tim50-GFP. This should be done under normal conditions, where it is shown that YFP-CL1 is distributed in the cytosol, but it is expected that around 20% of the protein would be linked to the mitochondria, as suggested by the authors' results.

Authors: We did not make any quantitative statements about the amount of YFP-CL1 that is associated with membranes. Having said that, based on our data it is reasonable to assume that a considerable amount of YFP-CL1 is bound to mitochondria. We have performed immunostaining which indeed confirms that there is co-localization between YFP-CL1 and TOM20. As a negative control, we included the soluble reporter Ub^{G76V}-YFP that is not subject to enhanced degradation in eIF5A-deficient cells and does not show an immunostaining consistent with binding to mitochondria.

See Fig 6a and Suppl. Fig. 7a and lines 347-350.

1.2 Suppl. Figure 1B: Treatment with expomicin/TAK-243 results in increased YFP-CL1 fluorescence and the formation of perinuclear foci. However, it is surprising that the authors do not present any experiments demonstrating the colocalisation of this fluorescent signal with mitochondria, which they propose as a cellular compartment for holding these aggregates. Including such colocalisation data would strengthen their argument.

Authors: It has not been our intention to suggest that the mitochondria are a holding compartment for protein aggregates. If we gave that impression, we apologize for the confusion. We instead argued that interaction between YFP-CL1 and mitochondria hinders efficient clearance and increases the steady-state levels of YFP-CL1. As the various pools of YFP-CL1 (soluble, ER-associated and mitochondria-associated) are likely to be dynamic, aggregation does not necessarily have to occur at mitochondria. Nevertheless, this is an interesting question and we have performed immunostainings for mitochondria in proteasome inhibitor-treated YFP-CL1 cells. This indeed shows that YFP-CL1 aggregates and mitochondria largely co-localize in the same subcellular region. We show as a negative control proteasome inhibitor-treated Ub^{G76V}-YFP cells. Also mHtt aggregates are observed in proximity to mitochondria. However, we refrain from proposing this to be a general phenomenon for the reasons outlined above.

See Suppl. Fig. 7f and Suppl. Fig. 10b and lines 372-375 and 445-446.

1.3 The localisation of the YFP-CL1 construct to mitochondria should also be investigated under conditions of eIF5A inhibition and mitochondrial dysfunction (OXPHOS inhibitors). Visualising mitochondria with a protein-independent marker such as Mitotracker (in addition to Tom20 localisation) is of interest, given that mitochondrial protein mislocalisation and aggregation in the cytoplasm have been reported under eIF5A depletion.

Authors: We included micrographs of control and GC7-treated YFP-CL1 cells stained with MitoTracker.

See Suppl. Fig. 9g and lines 403-405.

1.4 Controls using other reporters fused to different amphipathic helices should be employed to confirm the general role of mitochondrial attachment and eIF5A function in preventing the degradation of these aggregation-prone targets. One interesting reporter would be an endogenous cytosolic protein fused to CL1 and bearing an HA or myc epitope, as well as YFP fused to the htt amphipathic helix. This would allow protein levels to be determined, with direct or indirect immunofluorescent microscopy being used to study the localisation of the fused proteins.

Alternatively, subcellular fractionation could be employed to investigate the involvement of mitochondrial attachment and eIF5A in the regulation of protein stability.

Authors: To challenge our model, we turned to another disease-associated protein that contains amphipathic helices and has been found to be associated with mitochondria: mutant α -synuclein. We show that GC7 and BtdCPU treatments result in a decrease in the steady state levels of α -synuclein. Thus, in the revision we show for three unrelated amphipathic helix-containing proteins that the integrity of the mitochondrial network influences their steady-state levels.

See Fig. 8f,g and lines 467-477.

1.5 With respect the subcellular compartment fractionating shown in Fig.6, to conclude “YFP-CL1 was readily detected in both ER and mitochondrial fractions, indicating that the amphipathic helical CL1 interacts with both membranous compartments (Fig. 6A)”, it should be shown a control of YFP localization with the helix modified version, YFP-CL1*. In Fig. 6A, why is there less TIM50 in mitochondrial fraction when expression YFP-CL1??? Show Ponceau in Fig6A and also use the control TOM20. Additionally, it should be quantified the protein levels in Fig. 6A and B, with respect controls, from several experiments.

Authors: It has been well documented that the amphipathic helix of CL1 facilitates association with membranes as well as that the CL1* has reduced binding affinity (it has originally been designed and tested for this purpose). We expressed both YFP-CL1 and YFP-CL1* in cells, fractionated mitochondrial fractions and determined by western blotting the ratio of input to mitochondrial localization. In agreement with the data shown in the publication initially reporting the YFP-CL1* mutant, the amphipathic helix nature is strongly reduced, but not obliterated. This is in line with our data showing that only a minor pool of the YFP-CL1* can still bind to mitochondria compared to the original YFP-CL1. Regarding the fractionation experiments, we have carefully checked protein levels and secured equal loading of the YFP-CL1 reporter stable cell line. We would like to note that the TIM50 levels are very similar between the control and proteasome inhibitor-treated samples, which are the two critical samples for comparison whereas the parental cell line is only included to confirm the specificity of the YFP signal.

See Fig. 7d and lines 428-430.

In relation with this, in Fig. 5D and E it is shown NES-GFP-CL1 and NLS-GFP-CL1 reporters (the first affected and the second not by eIF5A), but in the text says: “we found that cytosolic localization of the reporter was a prerequisite for enhanced degradation induced by GC7 treatment (Fig. 5D), as the nuclear YFP-CL1 reporter was unresponsive (Fig. 5E)”. So, where is the mistake? Are GFP or YFP constructs used? This should be clarified and the experiment proposed above using different reporters should be done.

Authors: We thank the reviewer for pointing out this mistake. The compartment specific reporters are indeed based on GFP (gift from Ron Kopito, Stanford University) while our own home-made reporters are based on YFP. Note that GFP and YFP only differ in four amino acids. The GFP antibodies that are used in the study are commonly used for GFP or YFP detection. We should have stated “the nuclear GFP-CL1 reporter was unresponsive”. We have corrected this.

2. Additional experiments should be performed to exclude the possibility of an indirect effect of eIF5A mediated by a drop in cellular ATP levels caused by loss of mitochondrial activity under eIF5A depletion, and to discard a direct or indirect role of eIF5A in proteasome activity.

Authors: We are a bit puzzled by this question as both ubiquitination and proteasomal degradation are heavily dependent on ATP. It is well documented that depletion of ATP will cause a general inhibition of ubiquitin-dependent proteasomal degradation. Nevertheless, we have addressed a possible enhancing effect of eIF5A depletion on proteasome activity. Depletion or inhibition of eIF5A did not positively or negatively affect proteasome activity.

See Suppl. Fig. 5g,h and lines 312-315.

2.1 Experiments with ATP depletion in native cells and under low active eIF5A conditions should be done to inform of ATP dependent/independent role of eIF5A. It could be investigated the localization/stability of the reporter YFP-CL1.

Authors: For the reason outlined above, we have not tested the effect of ATP depletion on degradation of YFP-CL1. Note, that with two of the eIF5A siRNAs we detect still substantial oxidative phosphorylation, and inhibition of ATP synthase by oligomycin had similar effects on mitochondrial respiration as in the control. This is an additional indication that ATP depletion is unlikely to be responsible for the enhanced degradation in eIF5A-deficient cells.

See Suppl. Fig 8d.

2.2 Measure proteasomal activity with specific proteasomal degradation reporters in native cells and under eIF5A depletion, also with inhibitors of mitochondrial activity. This will allow to directly exclude an effect of eIF5A in proteasomal activity.

Authors: As stated above, we have tested proteasome activity and did not find an effect of eIF5A depletion or GC7 treatment.

See Suppl. Fig. 5g,h and lines 312-315.

2.3 eIF5A is supposed to be essential in all eukaryotic cells, so the isolation of mutants by CRISPR-Cas9 screening and also the subsequent isolation of eIF5A knockdown clones could carriage additional suppressor mutations. Therefore, one should be cautious and check most results with other ways of eIF5A inhibition. Experiments of loss of mitochondrial localization of YFP-CL1 (Fig.6B) should be also confirmed with depletion of eIF5A protein by using siRNA against eIF5A and/or GC7 treated cells.

Authors: As shown by us and others, eIF5A is not essential for eukaryotic cells (this study and <https://doi.org/10.1038/s41467-022-35252-y>). Deletion of eIF5A can result in a prolonged G1 phase and slowdown of cell growth but it is not lethal in cell culture. We agree that deletion of eIF5A may induce adaptive responses that may complicate the interpretation of the data. Therefore, we used two independent EIF5A knockout clones. We have isolated mitochondrial fractionations of GC7 treated cells and detected a reduction in YFP-CL1 reporter levels.

See Suppl. Fig. 7b and lines 361-362.

3. The eIF5A main role described is as translation elongation factor, therefore a direct role in the synthesis of YFP-CL1 must be ruled out.

3.1 In their system, the authors co-express YFP-CL1 and H2A-BFP in the same mRNA. Therefore, normalising the YFP signal against the BFP signal eliminates the effects of transcription. However, it cannot be assumed that the effects of translation/ protein synthesis are not affecting the results, since translation rates may differ between the two open reading frames (ORFs) under active eIF5A depletion. The authors demonstrate that blocking the proteasomal degradation of YFP-CL1 appears

to reverse the effects of eIF5A depletion. However, a more direct test is required to rule out the possibility of an eIF5A role in YFP-CL1 synthesis. To demonstrate that the effect of eIF5A is linked to stability rather than synthesis, it would be necessary to directly analyse YFP-CL1 protein stability in eIF5A-depleted cells (using siRNA against eIF5A and GC7 treatment) versus native conditions, employing a conventional translation shut-down experiment involving CHX and the detection of protein levels over time. This experiment should also be performed with the YFP-CL1* version with a modified amphipathic helix to control for any differences in levels and stability.

Authors: This point is well taken. We agree that expression of tBFP-H2A from the same transcript does not exclude the possibility that reduced synthesis is responsible for the reduced steady-state YFP-CL1 levels. Having said that, it would be hard to explain how eIF5A depletion only affects the synthesis of YFP-CL1 without reducing tBFP-H2A levels. Note that for a protein that has a half-life of approximately 30 minutes (like YFP-CL1), the predicted reduction in half-life to explain a 20% reduction in steady-state levels will be around 5 minutes. Even smaller differences are expected between YFP-CL1 and YFP-CL1*. Therefore, we relied in our paper on stabilization of the reporter in the presence of proteasome inhibitor as this not only confirms accelerated degradation but also identified the UPS as being responsible. We have performed the requested cycloheximide experiments and compared clearance of YFP-CL1 in parental and two EIF5A knockout cell lines. This showed that the EIF5A knockout cell lines cleared the YFP-CL1 faster.

See Suppl. Fig. 6c and lines 330-332.

3.2 From Fig. 3F it is stated that “depletion of eIF5A selectively reduced the steady-state levels of YFP-CL1 without a substantial decrease in the levels of ERAD and soluble substrates (Fig. 3F)”. However, if eIF5A plays a role in the translation elongation of the reporter, this may depend on the position of YFP within the open reading frame (ORF). On the one hand, it could be checked whether CL1-YFP level is affected by eIF5A inhibition; on the other hand, at least some of the ERAD and soluble reporters should be examined with YFP in the N-terminal part of the protein.

Authors: This is an interesting experiment. We generated a reporter with the CL1 positioned at the N-terminus of YFP. This resulted surprisingly in a more profound mitochondrial localization, which was accompanied by an increase in the steady-state levels and reduced proteasomal degradation, in line with mitochondrial localization hindering efficient degradation. In hindsight, it may not be that surprising that CL1 in the N-terminal position behaves as a mitochondrial localization signal given its amphipathic nature. The fact that high expression levels were obtained both in control and GC7-treated cells also argues against a role of eIF5A in translation of the CL1 sequence as it has been suggested that in particular N-terminal mitochondrial localization signals render proteins eIF5A dependent (<https://doi.org/10.1016/j.cmet.2019.05.003>).

See Suppl. Fig. 7d,e and lines 368-372.

4. The results with mutant htt are somewhat weak. In Figures 8D and 8E, the effect of proteasome inhibition on suppressing the effect of eIF5A is minimal. A similarly weak difference is reported when the amphipathic helix of htt is mutated.

4.1 We suggest probing the role of eIF5A in linking htt to mitochondria and regulating its stability through this mitochondrial attachment more directly. Biochemical fractionation and/or microscopy experiments should be conducted to determine whether htt is partially linked to mitochondria in an eIF5A-dependent manner, and whether this link depends on the htt amphipathic helix. The experiments should be quantified.

Authors: As mentioned in the manuscript, it has been previously shown that Htt localizes to mitochondria (<https://doi.org/10.1038/nn.3721>). We included now mitochondrial fractionations of mHtt-GFP expressing cells. Note that we also observed mHtt-GFP inclusions in proximity to mitochondria.

See Suppl. Fig. 10a,b and lines 441-446.

5. No molecular mechanism is suggested for the role of eIF5A in the mitochondria. Do the authors think that eIF5A facilitates the retention of the aggregation-prone peptides in the mitochondria? Is it helping the import of these peptides through the mitochondrial membranes?

5.1 To begin addressing this, co-immunoprecipitation experiments of eIF5A using mitochondrial fractions could provide insights. Additionally, more discussion of the potential eIF5A-mediated molecular mechanism would strengthen the manuscript and should be included in the Discussion section.

Authors: This is an important point. Although it is clear from previous work that eIF5A modulates mitochondrial homeostasis, the molecular mechanism responsible for this effect is less clear. It has been proposed that eIF5A is important for synthesis of proteins that contain an N-terminal mitochondrial localization signal (<https://doi.org/10.1016/j.cmet.2019.05.003>). Moreover, in yeast it has been proposed that eIF5A-dependent synthesis of TIM50 is critical for efficient import of mitochondrial proteins (<https://doi.org/10.1083/jcb.202404094>). We feel that this question is very interesting but lies outside the scope of the present study. In the revision, we refer to these potential mechanisms for the mode of action of eIF5A. We speculate in the model and text that the increased residence time at the mitochondria hinders efficient ubiquitination at the ER compartment.

See Fig. 9 and lines 506-509.

5.2 It has been recently shown that eIF5A depletion causes mitochondrial import defects because the stall of ribosomes in TIM50 mRNA, which encodes a protein of the MIM system. This defect in mitochondrial protein import to the mitochondria could cause the attachment of prone-aggregation proteins to the mitochondrial surface?? To address this, it could be examined the localization/stability/levels of YFP-CL1 and YFP-CL1* and/or mutant htt under conditions of depletion of Tim50 protein.

Authors: We are aware of this study. Importantly, human TIM50 does not contain the stretch of proline residues that render yeast eIF5A synthesis dependent on eIF5A (nor other eIF5A motifs). We feel that without any indications that the synthesis of human TIM50 is dependent on eIF5A, it would be premature to start to explore a role for TIM50 in the observed effect. We mention this now in the discussion that human TIM50 lacks the proline rich sequence.

See lines 546-550.

Other Major concerns:

6. The authors do not mention anywhere that eIF5A is encoded by two isoforms (EIF5A1 and EIF5A2) neither which isoform is targeted by the three silencing RNAs or deleted in the KO cell line. This distinction is important and should be clearly stated in the main text and figure legends throughout the manuscript.

Authors: This is an important point. We have mentioned this and refer throughout the text to their official name. Note that we refer to the official names for the genes encoding the human orthologues: eIF5A ([Gene: EIF5A \(ENSG00000132507\) - Summary - Homo sapiens - Ensembl](https://www.ncbi.nlm.nih.gov/gene/132507))

genome browser 115) and eIF5A2 (Gene: EIF5A2 (ENSG00000163577) - Summary - Homo sapiens - Ensembl genome browser 115). We included qPCR to analysis for eIF5A and eIF5A2 transcripts upon siRNA depletion and in the knockout cells.

See Suppl. Fig. 4a-c and Suppl. Fig. 6b and lines 261-267 and 321-323.

7. In this line, the authors do not mention anywhere which is the most relevant function of eIF5A in translation (relieving ribosome stalling at specific peptide motifs). This should be acknowledged in the main text.

Authors: We apologize for this and mention now in the concluding paragraph of the introduction this function of eIF5A.

8. Several experiments were performed using three individual silencing RNAs targeting eIF5A, yet none of them triggered a strong reduction in YFP-CL1 levels. Would it be possible to combine the three siRNAs simultaneously to better increase the knockdown efficiency and drive more pronounced effects on protein degradation? Has this been already tested? If this approach has already been tested, it should be mentioned and discussed.

Authors: Both western blotting and qPCR confirm that these siRNAs efficiently deplete eIF5A. Therefore, we do not feel that combining the siRNAs would be recommendable as it may also increase the risk of off-target effects.

9. Cell viability should be assessed following 24-hour treatments with different concentrations of GC7 to ensure that, although cell proliferation might be potentially reduced, the cells remain viable. This control is important to confirm that the observed effects are not simply due to cytotoxicity.

Authors: We have monitored the growth of parental and eIF5A-deficient cells as well as cells +/- GC7 treatment and included the latter in the manuscript.

See Suppl. Fig. 4g and lines 284-286.

Minor issues/suggestions:

1. Keywords suggested: mitochondria, eIF5A

Authors: Keywords are included.

See line 16.

3. Figure 1B. Band quantification, replicates, error bars and statistics are missing.

Authors: We quantified the bands in Fig. 1B and added an additional figure where we quantitatively follow the decrease of YFP-CL1 and tBFP-H2A upon cycloheximide administration using flow cytometry.

See Fig. 1c and text 140-141.

4. Figure 2B and 3B. The software or tool used to achieve the GO representation should be better included in the figure legend.

Authors: The software is described and cited in the Methods section.

5. Figure 2D. Please include the concentration used for the siRNAs in the figure legend.

Authors: We added the concentration of siRNAs used in Fig. 2d.

6. Figures 2E,F. Statistics relative to control as well as individual dots for each replicate value are missing.

Authors: We included the individual points and the statistics of the measured bar graphs of the combined treatment in Fig. 2e,f.

8. Figure 5A. Statistics are missing.

Authors: We added statistics.

9. Figures 5B,C. Statistical comparisons between untreated and GC7 treatment are missing.

Authors: We added statistics.

10. Figure 5G. Change to a classic bar graph with individual dots for each replicate value to maintain the general graphic style of the paper.

Authors: We added individual points.

12. Log fold change: sometimes written as “lfc” and other times as “LFC”. Please unify.

Authors: We corrected it all to LFC.

13. In general, most of the figures show the signal for YFP fluorescence measured by flow cytometry, but it is not specified if this is calculated by normalizing this signal to the tBFP one. Please include this information in every figure legend.

Authors: This is now clarified in the figure legends.

14. In general, there are many experiments with a lot of replicate values for the control (more than 5 most of the cases) while 2 or 3 values for the rest of the conditions. At least, 3 independent values should be shown for each experiment and better use same number of replicas.

Authors: Two controls were included for each condition to ensure reliability of the control for fold-change calculations.

15. Line 255: “eIF5A depletion significantly reduced YFP-CL1 levels...” Include which is the percentage of signal decrease.

Authors: This is mentioned for each of the siRNAs in Fig 4a.

16. Line 311, reference 42. Add the following references where it is demonstrated the eIF5A induces a metabolic switch from oxidative phosphorylation to glycolysis (<https://doi.org/10.1002/1873-3468.14366> and <https://doi.org/10.3390/ijms22010219>).

Authors: These citations have been included.

17. Line 329. “And the mitochondrial compartment interfering with this process.” Suggestion to add that this is also mediated by the action of eIF5A.

Authors: Added. Lines 364-365.

18. Line 389: “N-mut htt-GFP inclusions measured by flow cytometry”. Explain a little bit more in detail about how this is performed.

Authors: We described the method in more detail and included a citation to the original paper describing the PulSa method.

19. Please, add some information about Hoechst staining in the figure legends.

Authors: The information on the Hoechst staining (brand, dilution, incubation time, reagent identifier) is included in the Method section.

20. Figure 7B. Legend only specifies anti-GFP antibody, is anti-GFP or YFP?? Please, add details corresponding to other antibodies used here.

Authors: Throughout the text anti-GFP antibodies have been used. Since YFP and GFP only differ at four amino acids, anti-GFP antibodies are typically used for detection of YFP and other GFP variants.

21. Figure 7D. Add control images with no proteasome inhibition.

Authors: We added the control conditions in the figure. See Fig. 7e.

22. Figure 8A. Legends says n=3 but clearly is higher.

Authors: Due to the variation in transient transfections, we used always three biological samples for the control conditions and two biological samples for each treated condition per replicate here.

23. Figure 8E. Statistical comparisons between untreated and GC7 mut htt are missing.

Authors: We added statistics.

24. Figure 9. Please include eIF5A protein in the model representation since it is already in the figure legend.

Authors: eIF5A has been included.

25. Materials and Methods, Western Blotting section. Add conditions of centrifugation steps. Include how protein quantification is performed as well as the amount of proteins loaded on SDS-PAGE.

Authors: We included the centrifugation steps and the protein quantification information.

26. Supplementary Table 3. "Reciepe" change to recipe.

Authors: This typo has been corrected.

27. As mentioned above in Fig.5D, F (NES-GFP)-CL1, (NLS-GFP)-CL1, It should be clarify whether GFP or YFP has been use and correct text and figure accordingly.

Authors: This has been corrected.

28. Fig5F figure legend "plotted for their tBFP versus YFP ratio", should it be the opposite?

Authors: Thanks for spotting this. Indeed, it should have been the other way around. It has been corrected.

29. Fig. 6E Y-axis is written "aspect ratio", it should be better described what is measured: length of the mitochondrial filaments.

Authors: We report the aspect ratio and not the filament length. This is now clarified in the methods section.

30. Supplementary Figure 6E, include control of cells with eIF5A depletion.

Authors: We included a GC7 treated condition which is from the same experiment as the 24h treated rotenone sample.

Reviewer #3 (Remarks to the Author):

Authors: We thank Marina Barba-Aliaga for the helpful comments.

Rebuttal 2nd revision NCOMMS-25-56601 “Mitochondria serve as a Holdout Compartment for Aggregation-Prone Proteins hindering Efficient Degradation”

Authors: We would like to thank the reviewers for their constructive feedback and giving us the opportunity to correct and improve the manuscript.

Reviewer #1: Proteinase K interpretation (Suppl. Fig. 7c, lines 365-368): The interpretation might be slightly overreaching. Could the authors either add other control blots or soften the language in lines 365-368 to acknowledge the protein could also reside in the intermembrane space if the outer membrane was compromised during isolation.

Authors: We agree that we cannot exclude the possibility that the reporter substrate reaches the intermembrane space. In acknowledgement of this we have toned down the interpretation.

See new lines 369-372.

Please check the reference update (line 48, ref 3): The authors stated in their rebuttal that "This citation has been replaced with a review from 2024," but the current reference in the manuscript remains: "Soto, C. Unfolding the role of protein misfolding in neurodegenerative diseases. Nat Rev Neurosci 4, 49-60, (2003)." Please update this citation.

Authors: We apologize for this mistake. In the revision we have now included the review: Koszla, O. & Solek, P. Misfolding and aggregation in neurodegenerative diseases: protein quality control machinery as potential therapeutic clearance pathways. Cell Commun Signal 22, 421, (2024).

See lines 46-49, 850-852.

Reviewer #2&3: The authors have convincingly replied to most of the raised concerns and also presented a large number of new experiments to support their statements. We only have one minor concern regarding concern 4.1, which arose in our previous revision. The author did not answer the question of whether N-Htt protein association with mitochondria is eIF5A-dependent, as they have shown for YFP-CL1. A similar experiment to the one shown in Fig. 6c would be required.

Authors: We isolated mitochondrial fractions from eIF5A knockout cells and parental cells expressing GFP-tagged mutant huntingtin. Our data show that the pool of mitochondrial associated mutant huntingtin is dramatically reduced in the EIF5A knockout cells. This is consistent with our model that reduced mitochondrial association of aggregation-prone proteins is responsible for the enhanced degradation in eIF5A deficient cells.

See new Fig. 8f,g and lines 473-476, 1213-1219.